# Surface chemistry-mediated modulation of adsorbed albumin folding state specifies nanocarrier clearance by distinct macrophage subsets

Michael P. Vincent[1], Sharan Bobbala[1], Nicholas B. Karabin[1], Molly Frey[2], Yugang Liu [1], Justin O. Navidzadeh[1], Trevor Stack[1] & Evan A. Scott [1,2,3,4,5✉]

Controlling nanocarrier interactions with the immune system requires a thorough understanding of the surface properties that modulate protein adsorption in biological fluids, since the resulting protein corona redefines cellular interactions with nanocarrier surfaces. Albumin is initially one of the dominant proteins to adsorb to nanocarrier surfaces, a process that is considered benign or beneficial by minimizing opsonization or inflammation. Here, we demonstrate the surface chemistry of a model nanocarrier can be engineered to stabilize or denature the three-dimensional conformation of adsorbed albumin, which respectively promotes evasion or non-specific clearance in vivo. Interestingly, certain common chemistries that have long been considered to convey stealth properties denature albumin to promote nanocarrier recognition by macrophage class A1 scavenger receptors, providing a means for their eventual removal from systemic circulation. We establish that the surface chemistry of nanocarriers can be specified to modulate adsorbed albumin structure and thereby tune clearance by macrophage scavenger receptors.

[1] Department of Biomedical Engineering, Northwestern University, Evanston, IL 60208, USA. [2] Interdisciplinary Biological Sciences, Northwestern University, Evanston, IL 60208, USA. [3] Chemistry of Life Processes Institute, Northwestern University, Evanston, IL 60208, USA. [4] Simpson Querrey Institute, Northwestern University, Chicago, IL 60611, USA. [5] Robert H. Lurie Comprehensive Cancer Center, Northwestern University, Chicago, IL 60611, USA. ✉email: evan.scott@northwestern.edu

Soft polymeric nanocarriers, broadly defined by their deformability and organic composition[1,2], are versatile platforms for controlled and targeted delivery of therapeutic payloads in vivo. Efficacy of this process can be enhanced by engineering nanocarrier surface properties to promote favorable interactions with desired target cell populations, while evading off-target cellular uptake and clearance by the mononuclear phagocyte system (MPS). The first step in nanocarrier development therefore involves designing ideal synthetic surface properties for the application of interest, paying close attention to design considerations that include vehicle shape/morphology, size, surface charge, and ligand display[3]. However, cells do not directly interact with this bare surface, as the nano/biointerface is obscured by coating of adsorbed protein[4–6]. This "protein corona" forms within microsecond to millisecond time scales upon exposure to protein-rich biological fluids and defines the true biological identity and biodistribution of nanocarriers[7].

Strategies are therefore needed to modulate protein corona formation and direct the corresponding biological responses and therapeutic outcomes. Minimizing protein adsorption and/or controlling the composition of adsorbing protein corona are two commonly employed methods for improving nanocarrier performance. A standard strategy is to customize nanocarrier surfaces with a hydrophilic and/or inert polymer coating, most commonly polyethylene glycol (PEG), to minimize protein adsorption. PEG has been demonstrated to enhance nanocarrier circulation time by evading recognition by phagocytes of the innate immune system, most prominently macrophages (MΦ)[8,9].

Despite efforts to minimize protein adsorption, its occurrence is inevitable. This realization has shifted the focus of many investigations to understand the relationship between nanocarrier surface properties and the composition of the protein corona[10,11]. These compositional differences reveal trends that help predict nanocarrier circulation time and other performance metrics. For example, minimizing nanocarrier interactions with complement protein C3 (and its C3b derivative) antagonizes opsonization to improve nanocarrier circulation time, whereas nanocarriers that increase C3 abundance in the protein corona are cleared more rapidly. Compositional trends, such as minimizing complement activation[12], can be harnessed to design surfaces that minimize nanocarrier interactions with adsorbing protein species that have well-defined negative consequences on nanocarrier performance.

For intravenously (IV) administered nanocarriers, serum albumin usually accounts for a significant fraction of the protein coating formed on the underlying nanocarrier chassis, as it is the highest concentration protein in the blood. Despite this high fractional composition within adsorbed protein layers, albumin is considered to be relatively inert, as albumin-coated liposomes have been shown to reduce nanocarrier recognition by MΦ to an extent that was comparable to PEGylated liposomes[13]. Many proteins of lower blood concentration, such as Factor XII[14] and kininogen[15], have higher surface activity and are linked to key biochemical responses to nanocarriers such as complement activation and thrombosis. It is commonly thought that albumin improves nanocarrier performance by conferring stealth-like properties, as it does not possess solvent-accessible amino acid sequence motifs or folded domains that interact directly with cell receptors or complement proteins. Yet, the biological consequence of an adsorbing protein species is not only dependent on its presence or absence in the corona but depends also on the three-dimensional conformation of that protein in the adsorbed state. Many nanomaterials modulate adsorbed protein structure[16–19], but the role played by adsorbed protein folding state is often overlooked when evaluating nanomaterial performance.

Structurally altered forms of albumin are non-canonical ligands for various cell membrane glycoprotein receptors, including gp18 and gp30 receptors[20], as well as the MΦ class A1 scavenger receptor (SR-A1)[21,22]. In humans and mice, SR-A1 is highly expressed by tissue-resident MΦ[23], but the role of these receptors in recognizing and clearing soft nanocarriers has not been examined as a mechanism employed by the innate immune system. These observations motivate a body of questions that are the focus of the present study.

Here we examined whether surface principles established using gold, silica, and polystyrene solid core hard nanoparticles (NPs) for controlling adsorbed protein structure apply to the design of a self-assembled soft polymeric vesicle. Furthermore, we assessed whether controlling the structure of adsorbed albumin using surface chemistry could tune the stealth properties of an albumin-rich protein corona. And finally, we assessed the relevance of SR-A1-mediated recognition for nanocarrier capture by the MPS. To achieve these objectives, we modulated the structure of albumin within an adsorbed serum protein corona using simple and commonly employed chemical moieties, which we displayed at the surface of self-assembled polymeric nanocarriers, and examined whether MΦ SR-A1 functions as an innate surveillance mechanism to recognize denatured albumin at the nano/biointerface. We utilized polymeric vesicles, polymersomes (PSs), self-assembled from poly(ethylene glycol)-b-poly(propylene sulfide) (PEG-b-PPS) diblock copolymers as a model system possessing surface-displayed hydrophilic PEG blocks terminated with chemical groups of different charge and polarity. PEG-b-PPS nanocarriers are versatile non-inflammatory[24] drug delivery vehicles that are frequently employed for intracellular delivery[25] in response to cell-derived reactive oxygen species and/or stimuli-responsive oxidizing agents[26]. This study has significant implications for the rational design of nanocarriers to control protein-mediated interactions with immune cell subpopulations. The mechanisms established here will be particularly useful for tuning nanocarrier evasion or recognition by diverse MΦ populations in drug delivery and imaging applications.

## Results

**Self-assembly of PEG-b-PPS PSs and physicochemical characterization.** In this study, we employ PEG-b-PPS vesicles as model soft nanocarriers by varying their surface chemistry without modifying their nanostructure. Polymer variants were synthesized to terminate the PEG hydrophilic block with commonly employed functional moieties for PEGylated nanocarriers: methoxy (MeO, -OCH$_3$) and hydroxyl (OH, -OH); as well as a common biological functionality: phosphate (Phos, -PO$_4$) found in RNA, DNA, and cell membranes (Fig. 1a, Supplementary Table 1, and Supplementary Figs. 1 and 2). PS nanocarriers[24,25,27,28] were self-assembled from PEG-b-PPS diblock copolymers using flash nanoprecipitation (FNP)[29–32] (Supplementary Figs. 1–3). Representative cryogenic transmission electron microscopy (Cryo-TEM) images verified a vesicular morphology for each PS, with the dark opaque PPS hydrophobic layer separating the inner hydrophilic lumen (PEG-enclosed) from the outer PEG corona (Fig. 1b). Nanocarrier size and zeta potential were characterized using dynamic light scattering (DLS) and electrophoretic light scattering (ELS), respectively (Table 1), and no significant differences were found between the diameters for the three PS types (Supplementary Fig. 4). Vesicle morphology was further confirmed and characterized by small angle x-ray scattering (SAXS) using synchrotron radiation (Fig. 1c). A spherical vesicle model[33] (Eq. 2; see "Methods") was fit to the scattering profiles collected for Phos PS (Fig. 1d), OH PS (Fig. 1e), and MeO PS (Fig. 1f). The fit vesicle diameter was in good agreement with diameter measurements obtained by DLS for each PS type (Table 1 and Supplementary Table 2). Anionic Phos

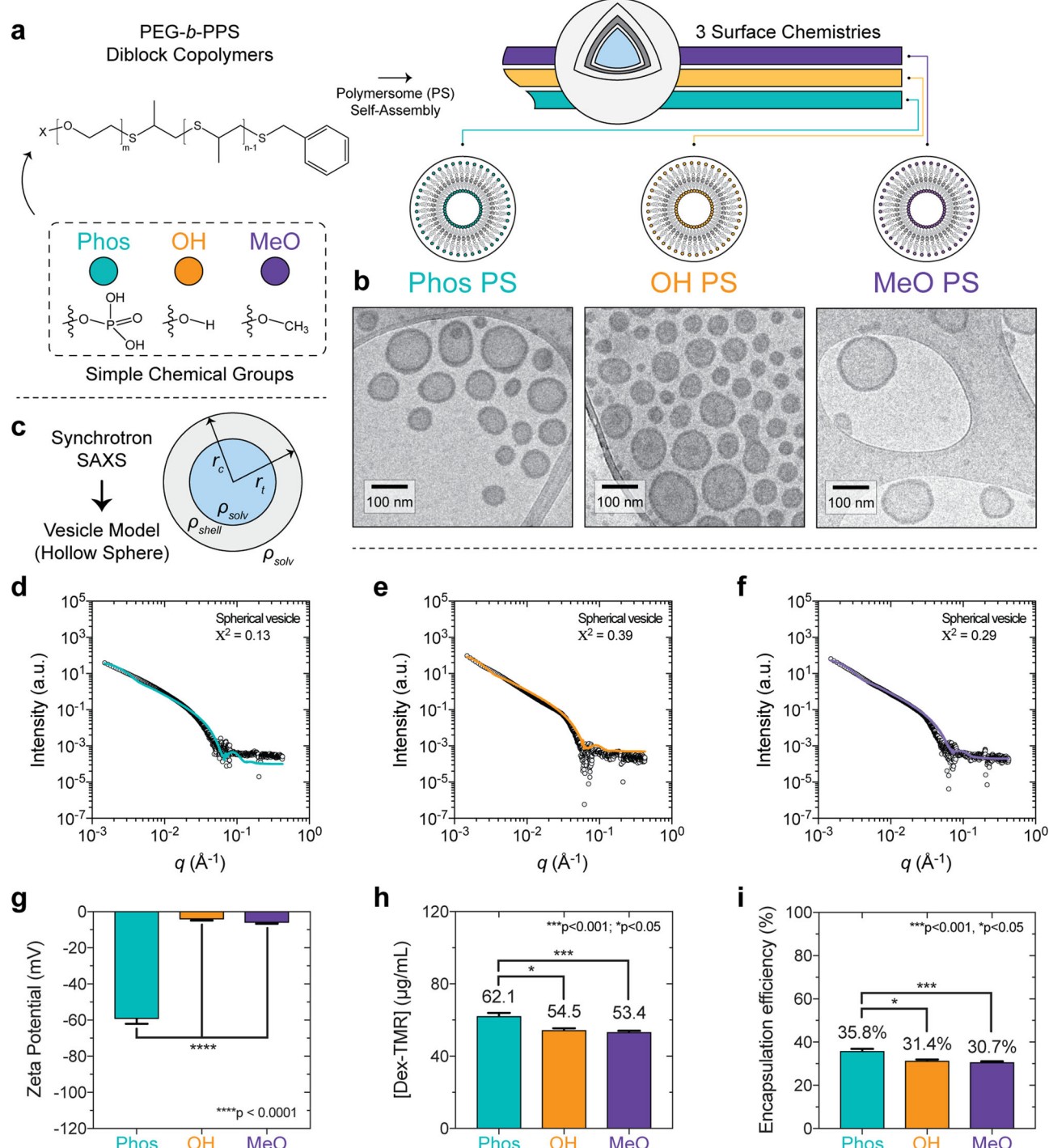

**Fig. 1 PEG-*b*-PPS polymersome morphology and hydrophilic loading characterization. a** Illustration depicting the poly(ethylene glycol)-*b*-poly(propylene sulfide) (PEG-*b*-PPS) polymersomes (PS) of three different surface chemistries. **b** Cryo-TEM of Phos PS, OH PS, and MeO PS. TEM micrographs were acquired at ×10,000 magnification (scale = 100 nm). **c** Illustration of PS characterization by small angle x-ray scattering (SAXS). A schematic of the spherical vesicle model is shown (core radius, $r_c$; total radius, $r_t$; scattering length density of the solvent and shell, $\rho_{solv}$ and $\rho_{shell}$, respectively). **d–f** SAXS scattering profiles and fit models for blank **d** Phos PS, **e** OH PS, and **f** MeO PS nanocarriers. SAXS was performed using synchrotron radiation. In each case, the scattering profile (gray dots) was fit using a spherical vesicle model. The model fit is represented as a solid line. The intensity (arbitrary units; a.u.) is plotted with the scattering vector ($q$). The chi square ($X^2$) for each model fit is displayed. A $X^2$ value of <1.0 indicates a reasonable model fit. **g** Zeta potential (mV) of Phos PS, OH PS, and MeO PS. The mean ± s.d. (n = 3) is displayed. **h, i**, Loading of the high molecular weight hydrophilic cargo. **h** The concentration of encapsulated 70 kDa Dextran-TMR (Dex-TMR) after purifying PS with size exclusion chromatography. **i** Dex-TMR encapsulation efficiency. For **h, i**, error bars represent s.e.m. from three parallel experiments (n = 3). In all cases, significance was determined by ANOVA with post hoc Tukey's multiple comparisons test (5% significance level). *$p < 0.05$, ***$p < 0.001$, ****$p < 0.0001$.

**Table 1 Physicochemical characterization of PEG-b-PPS polymersome formulations.**

| Nanocarrier | DLS | | SAXS | Charge |
|---|---|---|---|---|
| | Hydrodynamic diameter[a] (nm) | PDI[b] | Vesicle diameter[c] (nm) | Zeta potential[d] (mV) |
| Phos PS | 81.8 | 0.17 | 76.4 | −59.3 ± 1.6 |
| OH PS | 109.6 | 0.14 | 110.2 | −4.3 ± 0.6 |
| MeO PS | 98.3 | 0.13 | 99.8 | −6.2 ± 0.5 |

[a]Number average diameter measured by DLS.
[b]PDI (polydispersity index) calculated from the number average diameter distribution.
[c]Diameter values obtained after fitting SAXS profiles using a spherical vesicle model.
[d]Error represents the s.d.

PS had a significantly greater negative surface charge than OH PS and MeO PS as expected for their differences in pKa (Fig. 1g and Table 1). Our rheological characterization demonstrates PEG-b-PPS PSs to have an elastic modulus of <10 Pa (Supplementary Fig. 5a, b). In combination with their polymeric composition, this low modulus verifies the expected physicochemical properties of PEG-b-PPS PSs as soft NPs[1,2], distinguishing them from hard solid-core NPs such as gold, silica, and polystyrene that typically have an elastic modulus in the GPa range[1,34,35].

Encapsulation of a model hydrophilic dye cargo, 70 kDa dextran-tetramethylrhodamine (Dex-TMR), further verified the presence of inner aqueous lumens (Fig. 1h, i; see Supplementary Fig. 6 for calibration). The loading of Dex-TMR did not lead to major differences in the PS diameter or zeta potential (Supplementary Table 3). However, the loading of Dex-TMR did decrease the zeta potential of Phos PS to −50.3 ± 3.0 mV and increased the average diameter of OH PS to 123.2 nm (Supplementary Table 3). Despite decreases in cell viability observed after 8 h incubation with Phos PS and OH PS at high concentration, the Phos PS, OH PS, and MeO PS were all non-toxic to RAW 264.7 MΦ in the polymer concentration range of 0.01–1.0 mg/mL (Supplementary Fig. 7). Similar encapsulation efficiencies of ~30–36% were observed for all PS types (Fig. 1i and Supplementary Fig. 8a). Differences in the stability of cargo loading were observed after cold storage at 4 °C for 6 weeks (Supplementary Fig. 8a–c). The anionic Phos PS exhibited the greatest retention of Dex-TMR cargo, whereas the storage stability of Dex-TMR-loaded OH PS and MeO PS was lower (Supplementary Fig. 8b, c). In summary, our physicochemical characterization demonstrates that these PS are vesicular structures that differ in their surface chemistry and charge but not in size.

**Diverse MΦ populations are sensitive to PS surface chemistry in vivo.** We next examined whether surface chemistry differences can modulate the organ- and cellular-level biodistribution of PS in vivo in a mouse model. We hypothesized that PS surface chemistry is sufficient to alter preferential nanocarrier uptake by phagocytes of the MPS. Short time scales were of particular interest to us, since early interactions with phagocytes have major consequences on nanocarrier fate and overall performance in many applications. PS loaded with a near infrared dye, DiR, or phosphate-buffered saline (PBS) control were administered IV into C57BL/6J mice (four treatment groups), and organs were harvested after 4 h to assess early biodistribution differences (Fig. 2a). Whole-body organ perfusions were performed prior to organ dissection to minimize residual blood in the vasculature.

Each PS type was detected at comparable levels in the blood after 4 h (Fig. 2b). To estimate nanocarrier blood levels, we implemented a fluorescence assay that compares the DiR PS fluorescence in mouse plasma at 4 h to a physiologically relevant dilution of the input DiR-loaded PS prepared in untreated mouse plasma. This dilution serves as a proxy for nanocarrier blood levels at $t_0$ and theoretically represents the fluorescence of evenly dispersed DiR PS in the blood if no cellular uptake were to occur. This assay estimates that >55% of each formulation had been cleared from the blood by the 4 h timepoint (Fig. 2b). Since the majority of each formulation had already been cleared from the blood after 4 h (Fig. 2b), we reasoned that this timepoint was sufficient to provide a detailed picture of the organ- and cellular-level interactions taking place.

Our past assessment of the organ-level biodistribution of MeO PS detected high levels of these nanocarriers in the spleen, liver, kidneys, and lung as early as 1 h post-injection[36]. PS were also detected in the heart, albeit at lower levels[36]. Consistent with these data, at 4 h we detected PS of each surface chemistry in the spleen, liver, kidneys, lungs, and heart (Fig. 2c–h and Supplementary Fig. 9a, b). This is indicated by the significantly greater adjusted radiant efficiency determined in organs dissected from the PS treatment groups compared to those harvested from control mice receiving an IV injection of PBS (Fig. 2e–h and Supplementary Fig. 9b). A large fraction of each PS type was cleared by phagocytes in the spleen and liver (Fig. 2d–f). The kidneys and liver took up significantly less Phos PS nanocarriers compared to OH PS and MeO PS (Fig. 2d, g).

The results from our organ-level biodistribution analysis led us to question whether various immune cell subpopulations, especially MΦ, are sensitive to PS surface chemistry. To investigate this further, we prepared single-cell suspensions from each dissected organ and performed a flow cytometric analysis of PS uptake by a comprehensive set of immune cells (CD45+), as well as the pooled population of CD45− cells (Supplementary Tables 4 and 5).

Filtration in the spleen begins with the inflow of blood into the MΦ-lined marginal sinus. In the spleen, we assessed NP uptake by two MΦ populations, dendritic cells (DCs), two monocyte populations, and the pooled population of neutrophils and eosinophils. Splenic DCs preferred MeO PS over both of the polar surfaces (Fig. 2i). This is consistent with our past studies that demonstrated high DC uptake of MeO PS[36,37].

Our splenic cellular distribution analysis further examined nanocarrier uptake by (i) marginal metallophilic MΦ, which are referred to here as CD169+ MΦ due to their expression of CD169 (also known as Siglec-1 or Sialoadhesin)[38], as well as (ii) the CD11b+ F4/80+ MΦ (denoted by F4/80+ MΦ for simplicity). CD169 is a member of the immunoglobulin (Ig) superfamily that has evolved to have 17 Ig domains[39] and is conserved between humans and mice[40]. CD169-expressing MΦ have been implicated in exosomal capture in the spleen and lymph nodes (LNs)[41]. Past studies using albumin conjugated to CD169-targeting antibodies led to increased albumin internalization, confirming the involvement of CD169 in receptor-mediated endocytosis[42]. CD11b+ F4/80+ MΦ took up less MeO PS compared to Phos PS and OH PS surfaces (Fig. 2i). Interestingly,

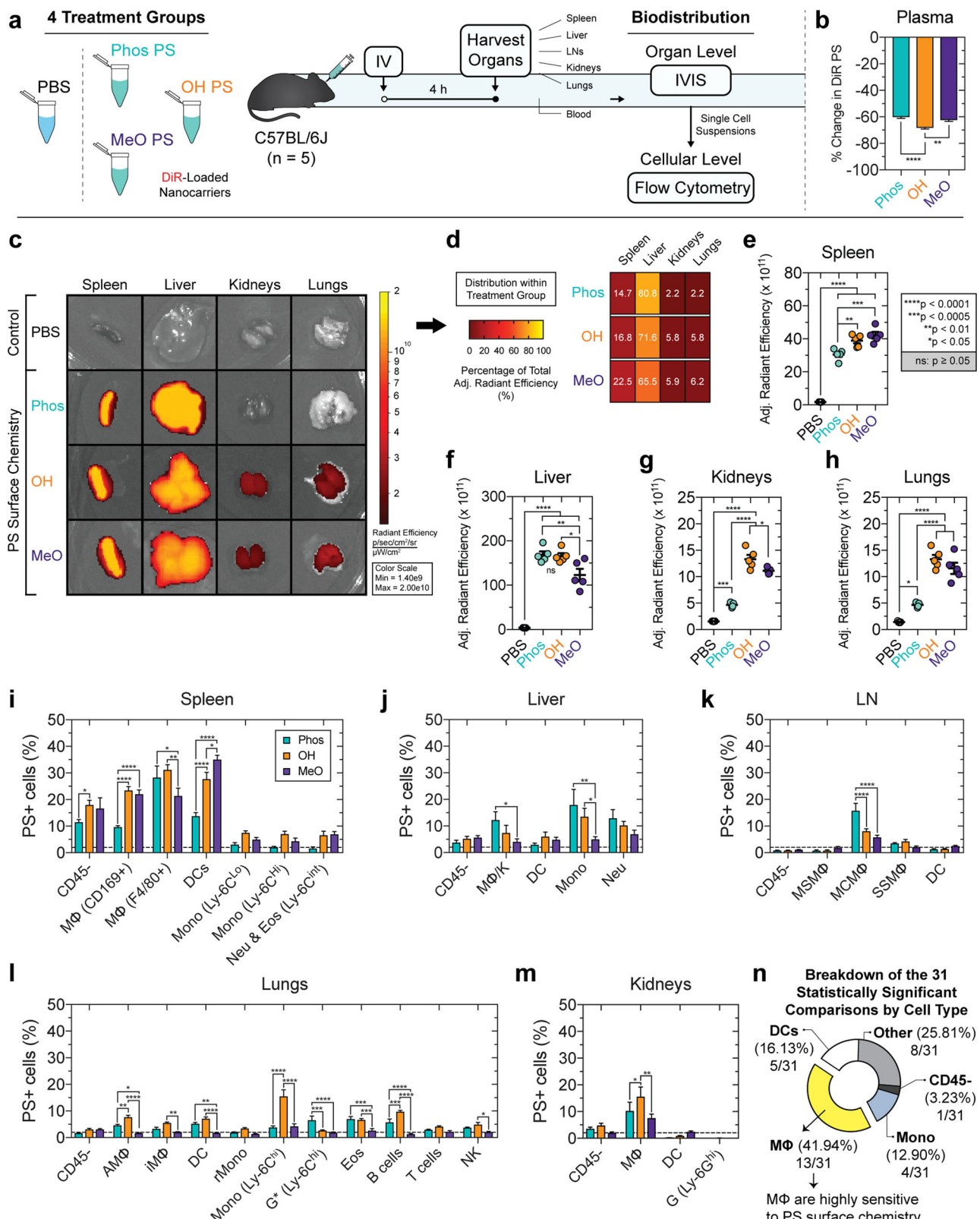

CD169+ MΦ exhibited a greater preference for MeO PS and OH PS than for Phos PS (Fig. 2i).

Liver MΦ (MΦ/K) and monocytes were sensitive to surface chemistry differences, whereas PS uptake by hepatic CD45− cells, DCs, and neutrophils was not (Fig. 2j). Hepatic MΦ/K and monocytes exhibited the strongest preference for the anionic Phos PS compared to either of the neutral PS. The liver was the lone

organ in our study that demonstrated a consistent preference for Phos PS uptake by multiple immune cell types.

Given the IV route of administration and the 4 h timepoint, we did not anticipate high levels of PS uptake by LN phagocytes. Nevertheless, we assessed PS uptake by LN CD45− cells, medullary sinus MΦ, medullary cord MΦ (MCMΦ), subcapsular sinus MΦ, and DCs (Fig. 2k and Supplementary Table 4).

**Fig. 2 Diverse murine macrophage populations are sensitive to polymersome surface chemistry in vivo. a** Overview of nanocarrier biodistribution study in four treatment groups of C57BL/6 J mice ($n = 5$) receiving intravenously administered PBS or near infrared dye, DiR-loaded PS (DiR PS). After 4 h, organ and cellular biodistribution was assessed by IVIS and multicolor flow cytometry, respectively. **b** Percent change in DiR PS fluorescence ($\lambda_{Ex} = 750$ nm; $\lambda_{Em} = 780$ nm) between serum (4 h) and a formulation-specific $t_0$ proxy. The $t_0$ proxy is the input DiR PS formulation diluted in untreated mouse plasma in a 1:15 ratio matching the ratio of DiR PS (100 μL) administered to the ~1.5 mL blood volume per mouse. **c–h** Organ biodistribution. **c** Representative IVIS images of organs. The radiant efficiency is displayed. **d** Distribution of total adjusted radiant efficiency within each treatment group, representing the percentage of the total adjusted radiant efficiency accounted for by the spleen, liver, kidneys, and lungs. **e–h** Adjusted radiant efficiency comparison between treatment groups in the **e** spleen, **f** liver, **g** kidneys, and **h** lungs. **i–m** Cellular biodistribution. The percentage of DiR PS+ cells of the specified type is displayed for the **i** spleen, **j** liver, **k** lymph nodes (LNs), **l** lungs, and **m** kidneys. Statistical significance was determined by ANOVA with post hoc Tukey's multiple comparisons test (5% significance level; *$p < 0.05$, **$p < 0.01$, ***$p < 0.0005$, ****$p < 0.0001$). Error bars represent s.e.m. **n** Percentage of the 31 statistically significant comparisons (annotated in **i–m**) accounted for by each cell type.

MCMΦ demonstrated a significant preference for Phos PS, with $15.8 \pm 2.8\%$ of MCMΦ cells being Phos PS+, compared to just $8.1 \pm 0.9\%$ and $5.8 \pm 0.7\%$ PS+ cells found for the OH PS and MeO PS treatment groups, respectively (Fig. 2k). MCMΦ involvement in the uptake of IV-administered nanocarriers is unsurprising, since these cells clear large blood-derived structures, such as apoptotic plasma cells[43,44].

We next examined an extensive panel of immune cells in the lung using a comprehensive gating strategy[45] with minor variations (Supplementary Table 5). Of the 11 lung cell types studied, 7 were sensitive to PS surface chemistry. Strikingly, five of these seven cellular populations exhibited a significant preference for OH PS (Fig. 2l). Inflammatory monocytes (Ly-6C^hi CD11c−, denoted by Mono (Ly-6C^hi)) were most sensitive to OH PS (Fig. 2l). We note that in this experiment, no treatment conditions were employed to stimulate monocyte recruitment (in any of the examined organs). But rather, these organ-associated monocytes are either tissue-resident monocytes present at some steady-state level (such as the lung-resident Ly6C− CD11c+ monocytes, denoted by rMono) or monocytes that are recruited on a short timescale (within the 4 h timepoint post-injection). It is unclear whether the inflammatory monocytes (Ly6C^hi CD11c−) were recruited or were present prior to nanocarrier administration. PS uptake by alveolar MΦ (AMΦ), interstitial MΦ (iMΦ), and B cells was also modulated by surface chemistry (Fig. 2l). The cellular biodistribution results in the lung suggest OH PS as a potential nanocarrier candidate for immunomodulation applications in the lungs, since it accumulates at higher levels than other surface chemistries in this organ (Fig. 2c, h), and is preferentially taken up by a variety of lung-resident immune cell subpopulations (Fig. 2l). OH PS is particularly well suited for delivering anti-inflammatory agents to combat inflammation-related respiratory conditions, owing to its preferential uptake by inflammatory monocytes, as well as the AMΦ and iMΦ populations.

Finally, PS uptake was examined for four cellular subpopulations in the kidney. Only renal MΦ were sensitive to PS surface chemistry (Fig. 2m). Statistically significant differences in uptake of each PS type were not found for renal CD45− cells, DCs, or granulocytes (Fig. 2m). The PS used in this study had an average diameter of >80 nm (Table 1), which exceeds size constraints for renal clearance[46].

Our results have significant implications on the rational design of nanocarrier surface chemistry to selectively target different immune cell subpopulations. The preferential uptake of OH PS and MeO PS by CD169-expressing MΦ in the spleen is particularly interesting. CD169 expression is largely restricted to MΦ, and targeting CD169+ MΦ has been achieved by displaying sialic acid-carrying ligands for CD169 at the surface of liposomes[47]. The preference for splenic CD169+ MΦ observed here suggests that PS with neutral surfaces are a useful chassis for immunomodulation applications targeting CD169+ MΦ. These applications include anti-inflammatory agent delivery to CD169+

inflammatory MΦ, such as those involved in rheumatoid arthritis[48], and vaccine applications in anti-cancer immunotherapy.

Collectively, our results demonstrate that MΦ are particularly sensitive to PS surface chemistry (Fig. 2n). Since PS introduction into the protein-rich blood results in rapid protein adsorption, we reasoned that differences in immune cell uptake preferences may be due to modulation of the protein corona and its consequence on cellular interactions, rather than differences in the bare synthetic surfaces that are covered by these proteins. Since the influence of opsonization processes on nanocarrier fate have been long established, we examined contributions mediated by adsorbed proteins that lack a direct link to a clearance-promoting function. We specifically assessed the role of serum albumin adsorption in increasing MΦ uptake. Albumin is the highest concentration protein in the blood and is one of the most abundant protein corona constituents, yet little is known regarding its negative consequences on nanocarrier fate.

**Albumin adsorption to PSs increases their uptake by MΦ.** Serum protein adsorption to each PS type results in the formation of a heterogeneous protein corona that is rich in serum albumin. Our in vivo studies demonstrated that >50% of each PS type has been cleared from the blood by 4 h following IV administration (Fig. 2b). We therefore investigated an even earlier timepoint of 2 h, which would both be relevant for these early clearance events as well as allow formation of a reasonably stable albumin-rich protein corona to minimize confounding effects arising from the potential dynamic exchange of albumin at later timepoints. Furthermore, we employed label-free quantitative mass spectrometry (MS) to demonstrate that serum albumin is a major constituent of the Phos PS, OH PS, and MeO PS protein corona formed in pooled human plasma at 2 h and 24 h timepoints in a separate study[49]. Consistent with these findings, serum albumin was also a major constituent of each protein corona formed in fetal bovine serum (FBS), which is indicated by the ~67 kDa band observed on silver-stained sodium dodecyl sulfate–polyacrylamide gel electrophoresis (SDS-PAGE) gels[50] (Supplementary Fig. 10). To examine the effect of adsorbed albumin on PS uptake by MΦ, we formed a protein corona using purified albumin, and we first examined the concentration dependence of albumin adsorption to each PS chemistry (Fig. 3a), the relationship between the albumin adsorption level and electrostatic changes, and albumin corona-mediated changes in PS uptake by MΦ (Fig. 3b–e). Albumin is a conserved heart-shaped transport protein consisting of three domains and carries a negative charge at physiological pH[51,52] (Fig. 3b). Fatty acid-free bovine serum albumin (BSA) was used in our studies, since this protein is inexpensive and there is little difference between serum albumin homologs in bovine, mouse, and human. This is demonstrated by a pairwise structural alignment performed on the α-carbon backbone of these serum

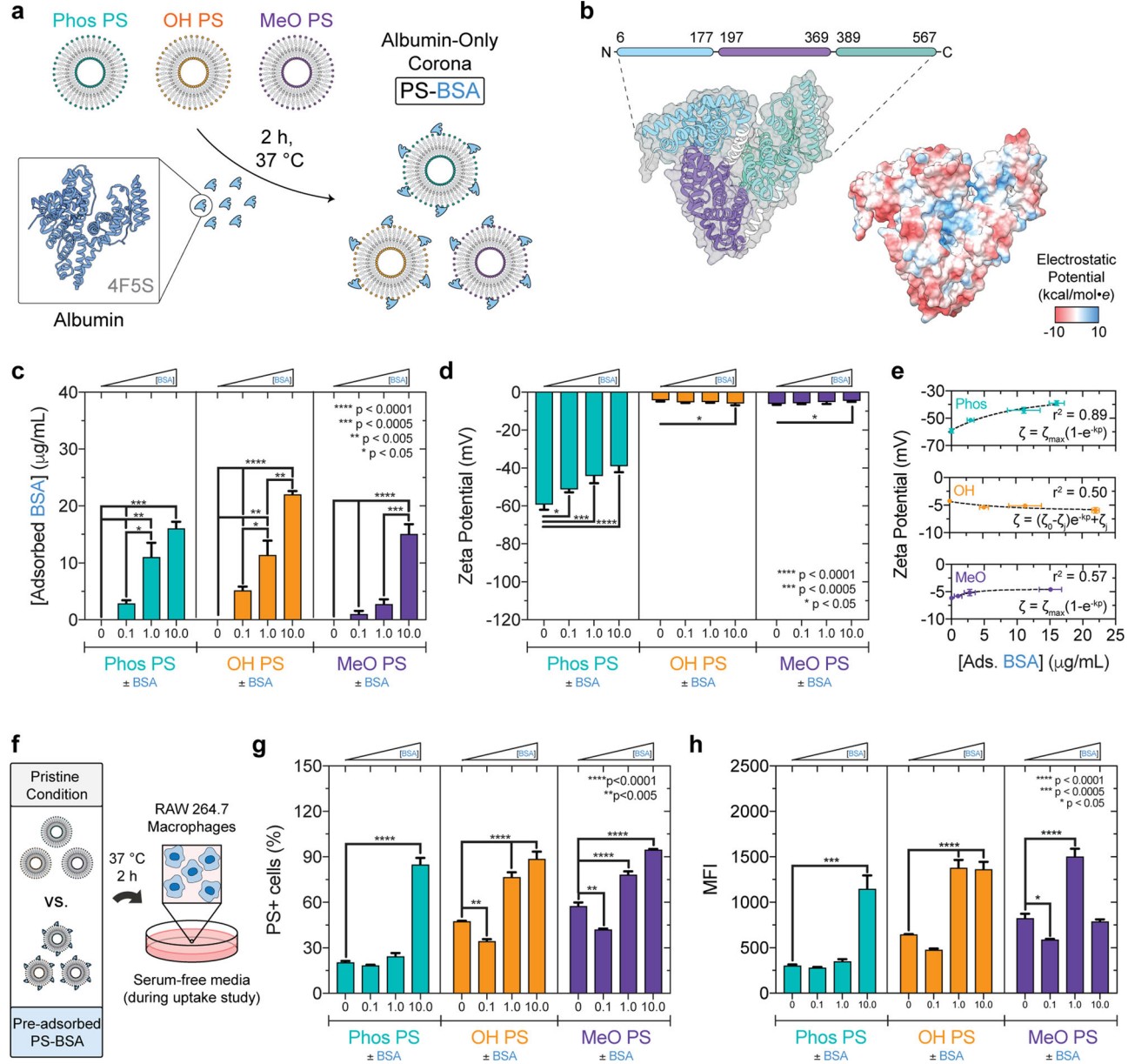

**Fig. 3 Adsorbed albumin increases macrophage uptake of all three PS surface chemistries but only significantly changes the zeta potential for Phos PS. a** PS–BSA complexes formed after polymersome (PS) incubation with bovine serum albumin (BSA; 0.1, 1.0, or 10.0 mg/mL). **b** Pfam domain annotations and Columbic electrostatic surface mapped to the BSA crystal structure (PDB: 4F5S). **c** Albumin adsorption concentration dependence (2.5 mg/mL PS; x-axis: input BSA concentration). Significance was determined by ANOVA with post hoc Tukey. **d** Zeta potential (mV) of PS and PS–BSA complexes. The mean ± s.d. is displayed from three parallel experiments ($n = 3$). **e** PS zeta potential as a function of adsorbed albumin. Non-linear fits (dashed lines) for one-phase association (Phos, MeO) and exponential decay (OH) models. **f–h** Flow cytometric analysis of PS–BSA uptake by RAW 264.7 macrophages in serum-free media (2 h, 37 °C). PS encapsulated Dex$_{70\,kDa}$-TMR to quantify vesicle uptake. **f** Illustration. **g** %PS+ cells (1.1% false positive rate). **h** Median fluorescence intensity (MFI). Error bars represent s.e.m. from three biological replicates ($n = 3$) unless indicated otherwise. Significant differences (PS versus PS–BSA) were determined by ANOVA with post hoc Dunnett's test. Statistical tests used a 5% significance level. *$p < 0.05$, **$p < 0.005$, ***$p < 0.0005$, ****$p < 0.0001$.

albumin homologs, which found an average root-mean-square deviation (RMSD) of ≤1.5 Å (Supplementary Fig. 11).

To assess whether albumin adsorption to PS increases with the concentration of albumin present during incubation and whether surface chemistry had a significant effect on the level of adsorbed protein, PS were incubated with a BSA concentration series (0–10 mg/mL) (Fig. 3c). The resulting PS–BSA complexes were isolated by ultracentrifugation and adsorbed protein concentration was measured using the Pierce A660 assay calibrated against a BSA concentration series (Fig. 3c and Supplementary Fig. 12a). PEG-

*b*-PPS polymers consisting of phosphate-, hydroxyl-, or methoxy-terminated PEG do not interfere with protein concentration determination by the Pierce A660 assay (Supplementary Fig. 12b). Albumin adsorption to PS increased in a concentration-dependent fashion (Fig. 3c), with both the PS surface chemistry ($p = 0.0001$) and the input BSA concentration ($p < 0.0001$) significantly affecting the level of BSA adsorption (Supplementary Table 6). For an experiment of this size, if PS surface chemistry has no effect on the adsorbed BSA concentration, there is a 0.01% chance of randomly observing an effect this large in magnitude

(Supplementary Table 6). The lowest level of total BSA adsorption was observed for the MeO PS surface, and this trend was found for all BSA input concentrations examined (Fig. 3c).

Charge shielding can result from protein adsorption to nanocarrier surfaces and alter their electrostatic interactions with physiological environments[53–56]. We therefore examined the relationship between albumin adsorption and changes in the zeta potential of PS surfaces. Albumin is negatively charged at physiological pH since its measured isoelectric point is ~4.9[57]. The zeta potential of PS and PS–BSA complexes was determined from ELS measurements. BSA adsorption to Phos PS resulted in significant concentration-dependent reductions in surface charge magnitude (Fig. 3d). This charge decrease, observed with increasing adsorbed albumin concentration, reasonably followed a simple one-phase association model ($r^2 = 0.89$; Fig. 3e). Albumin adsorption had much less of an impact on OH PS and MeO PS charge (Fig. 3d, e). Despite statistically significant differences observed in two cases, the magnitude of the changes was minimal compared to those observed for Phos PS.

We next assessed whether albumin adsorption alters PS uptake by RAW 264.7 MΦ (Fig. 3f–h and Supplementary Fig. 13a–c) and whether the observed changes in cellular uptake correlate with electrostatic changes. As most cells have a net negative surface charge, the rate of cellular uptake for anionic and neutral NPs is usually lower than that of cationic NPs due to electrostatic repulsion[56,58,59]. To examine the cellular uptake of nanocarriers, we incubated MΦ with nanocarriers for 2 h, washed cells with PBS, and measured the emission of nanocarrier-loaded fluorescent dye using flow cytometry. Our past work demonstrates that this general procedure quantifies fluorescence resulting from the cellular internalization of dye-loaded PEG-b-PPS nanocarriers, and the extensive washing procedure minimizes signal that would otherwise arise from nanocarriers associated with the plasma membrane (on the extracellular side) after the preceding media aspiration step[60–62]. Here this is further supported by lysosomal colocalization studies (Supplementary Fig. 14). After internalization, PEG-b-PPS PS traffic to lysosomal compartments and this colocalization is readily observable by confocal microscopy (Supplementary Fig. 14).

In separate studies performed using serum-free conditions, the baseline uptake of pristine PS (PS lacking a protein corona) was consistent with an electrostatic repulsion-mediated decrease in the cellular uptake of anionic nanocarriers (Fig. 3f–h, compare uptake of different PS types in the absence of adsorbed BSA; Supplementary Fig. 15). After pre-incubating washed cells in serum-free Dulbecco's modified Eagle's medium (DMEM) for 30 min, cells were incubated with Dex-TMR-loaded PS for an additional two-hour period prior to assessing uptake by flow cytometry (Fig. 3f). Serum-free conditions were used to enable the analysis of uptake differences arising from differential interactions between the cultured cells and the synthetic PS surfaces without interference from adsorbing protein species arising from FBS-supplemented media. The viability of RAW 264.7 MΦ is unaffected by serum-free media, even at a longer incubation period than used in our uptake studies (Supplementary Fig. 16). Compared to neutral OH PS and MeO PS, 264.7 MΦ took up significantly less anionic Phos PS after 2 h (Fig. 3g, h and Supplementary Fig. 1). Since these PS differed significantly in zeta potential and not morphology or size (Fig. 1 and Supplementary Fig. 4), we attribute the lower uptake of pristine Phos PS to greater electrostatic repulsion with the MΦ cell membrane.

Our analysis of adsorption-mediated electrostatic changes suggests that the external charge of Phos PS is altered most significantly by albumin (Fig. 3d, e). We therefore hypothesized albumin adsorption would have the greatest effect on the rate of Phos PS uptake by MΦ, since the highly significant reductions in

Phos PS negative charge (Fig. 3d) should decrease electrostatic repulsion with the negatively charged cell surface. To test this hypothesis, we examined the rate of PS nanocarrier uptake by MΦ in serum-free conditions after pre-incubation of PS with albumin at increasing concentrations (Fig. 3f). In most cases, albumin pre-adsorption increased the cellular uptake of PSs (Fig. 3h, i). The greatest protein-mediated increase in nanocarrier uptake was observed for Phos PS pre-incubated with 10 mg/mL albumin (Fig. 3g, h), which coincided with an albumin concentration of $16.1 \pm 1.2\,\mu g/mL$ adsorbed to the Phos PS surface (Fig. 3c) and a corresponding reduction in negative charge magnitude from $-59.3 \pm 2.8\,mV$ (pristine Phos PS) to $-39.1 \pm 3.2\,mV$ (Phos PS–BSA formed with 10 mg/mL BSA) (Table 1 and Fig. 3d). These results suggest that the albumin corona-induced decrease in negative surface charge for Phos PS lowered their electrostatic repulsion from the cell membrane to increase uptake (Fig. 3g, h). Electrostatic considerations do not explain the observed changes in MΦ uptake of OH PS and MeO PS following adsorption, since albumin adsorption resulted in minimal changes in the zeta potential of these nanocarriers (Fig. 3d).

In summary, these results demonstrate that albumin adsorption increases PS uptake by MΦ for each surface chemistry examined. Evidence for electrostatic contributions to this outcome was only found for Phos PS–BSA complexes. While albumin adsorption did modulate the MΦ uptake of neutral OH PS and MeO PS, a clear trend for electrostatic changes in mediating these effects was not established.

**Polymersome surface chemistry modulates the three-dimensional conformation of adsorbed albumin.** Tissue-resident MΦ express high levels of SR-A1[23] and structurally modified forms of albumin have been reported as non-canonical ligands for these receptors[21,63,64]. We reasoned that, if PS surface chemistry stabilizes or denatures albumin in the protein corona, then MΦ might utilize SR-A1 to sense these features on PS surfaces. We therefore characterized adsorption-mediated changes in the three-dimensional structure of albumin using a combination of biophysical and biochemical techniques (Fig. 4). Albumin is primarily composed of α-helical secondary structure elements[51,52,65]. We hypothesized the polar Phos PS and OH PS surfaces would preserve the native fold of albumin, whereas adsorption to the slightly more hydrophobic MeO PS would allow albumin to spread out and partially unfold to a certain degree. For clarity, we note here that, since albumin is natively folded prior to introducing the protein to nanocarrier suspensions, any observed loss of structure will be referred to as an unfolding or denaturation event in the present work.

A tryptophan quenching assay was used to examine whether MeO, OH, or Phos PS surfaces unfold adsorbed albumin (Fig. 4a). The basis of this assay is that the intrinsic fluorescence of a natively folded protein diminishes in the presence of a denaturant[66]. This principle allowed different levels of albumin unfolding to be distinguished due to increasing concentrations of the chaotrope guanidine hydrochloride (Gdn-HCl) (Fig. 4b)[67–69]. A significant ($p < 0.01$) decrease in albumin fluorescence is observed upon adsorption to OH PS, whereas changes in fluorescence upon adsorption to Phos PS or MeO PS surfaces were not statistically significant (Fig. 4c).

Limited proteolysis performed on albumin pre-incubated with Phos PS, OH PS, MeO PS, or a PBS control further verified that PS surface chemistry promotes different albumin conformations upon adsorption. Globular proteins, such as albumin, are less susceptible to proteolysis by trypsin in their natively folded state, whereas denatured forms of the protein are cleaved more readily

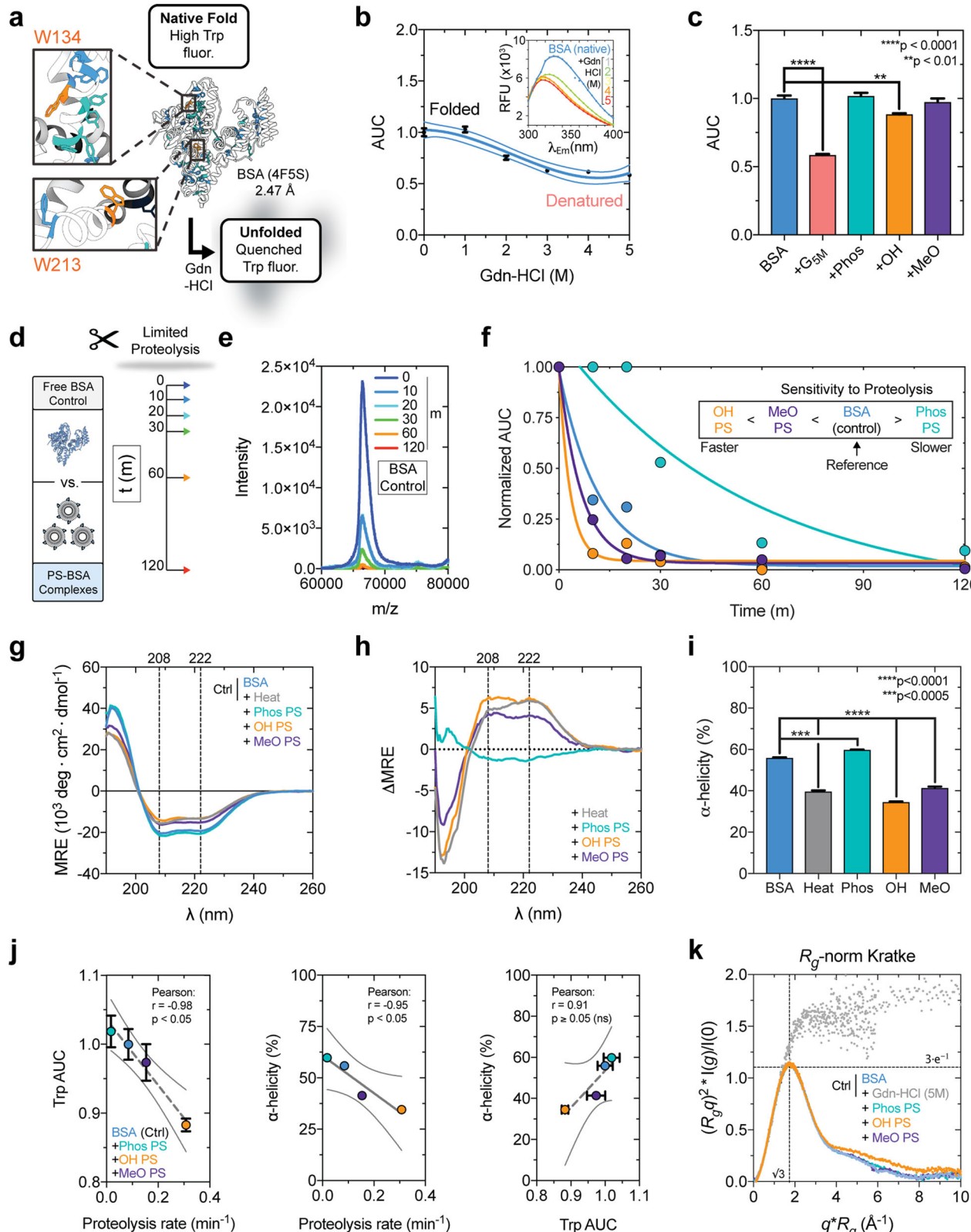

by protease[70,71]. Albumin proteolysis was quenched with formic acid at numerous timepoints over the course of 2 h (Fig. 4d). The time-dependent proteolysis of the full-length albumin control and albumin in the PS coronas was measured by matrix-assisted laser desorption time-of-flight MS (MALDI-TOF MS; Fig. 4e, f). For each condition, a one-phase decay model was fit to the $t_0$-normalized peak area plotted with time (Fig. 4f). The apparent

rate of proteolysis for free albumin ($\sim$0.09 min$^{-1}$) was much slower than for albumin adsorbed to OH PS ($\sim$0.31 min$^{-1}$) and MeO PS ($\sim$0.15 min$^{-1}$). The accelerated proteolysis observed for albumin adsorbed to OH PS and MeO PS further suggests that albumin partially unfolds upon adsorbing to these surfaces and that the hydroxylated (OH) surface more strongly denatures albumin than the methoxy (MeO) surface (Fig. 4f). On the other

**Fig. 4 Polymersome surface chemistry modulates the conformation of adsorbed albumin. a** Tryptophan-quenching assay. BSA (PDB: 4F5S) W134 and W213 residues (orange). **b** BSA fluorescence ($\lambda_{Ex} = 280$ nm) quenching after chaotropic guanidine hydrochloride (Gdn-HCl) denaturation. The AUC obtained by integrating emission spectra (inset) was normalized to folded BSA (third-order polynomial model ($r^2 = 0.90$), 95% confidence interval displayed). **c** BSA Trp quenching after adsorption to PS (Phos, OH, MeO); AUC normalized to folded BSA. $G_{5M}$ = Gdn-HCl-denatured BSA control. The mean ± s.e.m. is displayed from three parallel experiments ($n = 3$). Significant differences (versus folded BSA) were determined by ANOVA with post hoc Dunnett's test. **d**, **e** Limited proteolysis assessment of BSA cleavage susceptibility. **d** Trypsin proteolysis kinetics. **e** Full-length BSA cleavage with time monitored by MALDI-TOF MS (control shown). **f** Free BSA and PS-adsorbed BSA proteolysis kinetics ($t_0$-normalized). One phase decay models, $y = (y_0 - y_p) \times e^{-kx}$, were fit ($x$ = time (min); $y_0 = AUC_{t0}$; $y_p$ = AUC plateau; $k$ = proteolysis rate (min$^{-1}$)). Model $r^2$ values: 0.97 (BSA$_{control}$), 0.87 (Phos PS), 0.99 (OH PS), and 0.99 (MeO PS). **g**–**i** Circular dichroism (CD) spectroscopy of BSA, heat-denatured BSA control (85 °C, 10 min), and PS-adsorbed BSA. The **g** mean residue ellipticity (MRE) ($10^3$deg cm$^2$/dmol), **h** difference spectra w.r.t. folded BSA (PS–BSA$_{MRE}$ − BSA$_{MRE}$), and **i** α-helicity percentage. The mean ± s.e.m. from three parallel experiments is displayed ($n = 3$). α-Helicity significant differences (versus folded BSA) were determined by ANOVA with Dunnett's test. **j** Protein structure assay correlations (Pearson correlation coefficient ($r$), significance inset). Fit linear regression models are displayed with 95% confidence intervals. **k** $R_g$-normalized Kratke analysis of BSA SAXS profiles after adsorption. Error bars represent s.e.m. ($n = 3$). All statistical tests used a 5% significance level. **$p < 0.001$; ***$p < 0.005$; ****$p < 0.0001$.

hand, albumin adsorption to Phos PS resulted in a slower rate of proteolysis (~0.02 min$^{-1}$) compared to free albumin (Fig. 4f), which is consistent with albumin retaining its fold upon adsorption to the anionic Phos PS surface.

Collectively, our limited proteolysis results (Fig. 4f) are consistent with those from the tryptophan quenching assay (Fig. 4c). However, these assays merely suggest bulk structural changes upon adsorption. For a more detailed biophysical examination of adsorption-mediated changes in protein secondary structure, circular dichroism (CD) spectroscopy was performed to assess changes in the α-helix-rich content of the albumin corona. CD spectroscopy confirmed OH PS denatures albumin to the greatest extent (Fig. 4g–i and Supplementary Table 7). The differences in spectra revealed the level of denaturation to exceed that of 10-min heat denaturation at 85 °C (Fig. 4h) and corresponds to a highly significant ($p < 0.0001$) decrease in α-helicity to ~34.9 ± 0.4% (compared to 55.9 ± 0.3% measured for natively folded albumin) (Fig. 4i and Supplementary Table 7). MeO PS denatured albumin to a lesser extent, decreasing α-helicity to 41.4 ± 0.6%, whereas Phos PS slightly increased the α-helicity of albumin to 59.8 ± 0.2% (Fig. 4i and Supplementary Table 7).

Correlation analyses tie together each result from our structural analyses presented thus far (Fig. 4j). An increased protease sensitivity negatively correlates with normalized tryptophan fluorescence (Fig. 4j, left) and α-helicity (Fig. 4j, middle). Albumin is most strongly denatured by the OH PS surface and, as such, decreases in Trp fluorescence and α-helix content (Fig. 4j). The opposite was observed for Phos PS, which greatly stabilized the albumin fold (Fig. 4j). While α-helicity increased as a function of Trp fluorescence in our assays, the positive correlation was not significant but is attributable to the departure from linearity (Fig. 4j, right).

To further investigate conformational changes in albumin structure after adsorption to each PS type, we performed SAXS on PS–BSA complexes using synchrotron radiation (Supplementary Fig. 17). BSA was adsorbed to the surface of Phos PS, OH PS, and MeO PS, and differences in the scattering of high intensity 10 keV x-rays were assessed. The relative concentration of albumin and polymer was optimized to achieve a scattering intensity for BSA that falls within the dynamic range of measurement, while minimizing background scattering from the PSs.

To assess whether adsorption of albumin alters its apparent flexibility or symmetry, we converted the Kratke plot into dimensionless form normalized by the Guinier-estimated radius of gyration ($R_g$) (Fig. 4k). In this plot, compact globular proteins that obey Guinier's approximation exhibit a peak at √3 and of magnitude $3 \times e^{-1}$ (1.104). This expectation holds regardless of protein size or concentration. Departures from this waveform

indicate asymmetry or flexibility[72,73]. Chemically denatured albumin (control) exhibited a strong departure from the curve shape, peak, and magnitude, whereas albumin adsorbed to all PS surface chemistries exhibited a peak positioned at √3. In this study, PS–BSA did not exhibit substantial departures from the peak but instead exhibited departures in the tail region of the waveform. This observation suggests that the inherent flexibility and symmetry of albumin is not substantially altered upon adsorption to all three PS types, despite the potential partial unfolding in the case of PS OH.

Collectively, these data suggest that the structure of albumin is disturbed the least by the Phos PS surface and this anionic surface may stabilize the albumin fold (Fig. 4). Albumin appears to partially unfold upon adsorption to MeO PS and OH PS surfaces (Fig. 4). Surprisingly, adsorption to OH PS led to the greatest loss in albumin secondary structure (Fig. 4 and Supplementary Table 7). Our structural investigation found some similarities and differences from biophysical analyses of protein adsorption using polystyrene and silica NPs. Anionic carboxylate (COOH)-modified polystyrene NPs (60 nm diameter) stabilized albumin structure[19]. Our results with Phos PS are in agreement with this finding. Hydroxy-terminated (hydrophilic) spherical silica NPs preserve the structure of adsorbed albumin, whereas alkane-terminated (hydrophobic) silica spheres had a stronger denaturing effect[17]. This study used infrared spectroscopy to demonstrate that albumin adsorption to alkane-terminated hydrophobic silica spheres (≥70 nm in diameter) resulted in a loss of α-helical content down to 15–20%, with a corresponding increase in the fraction of the structure existing in a disordered random coil or extended chain state[17]. This same study demonstrated that hydroxy-terminated silica spheres of comparable size decreased α-helix content down to 25–30%[17].

Similarities aside, our results using PS self-assembled from PEG-b-PPS polymer show some differences from these past studies. For example, OH-functionalized PS denatured albumin to the greatest extent, even more so than the more hydrophobic MeO PS (Fig. 4). This result is in contrast to those reported with silica spheres, which demonstrated the non-polar hydrophobic surfaces to exhibit stronger denaturing effects than polar hydrophilic surfaces. The differences likely reflect inherent surface differences between silica NPs and PEGylated soft vesicles. The result from OH PS was particularly notable, since we initially hypothesized that this surface chemistry would stabilize albumin rather than denature it. This result suggests that a spherical arrangement of hydroxyl groups at the surface of a fluid PEGylated vesicle may promote albumin spreading in the absence of significant stabilizing anionic forces (such as in the case of the anionic Phos PS surface chemistry) that may interact with cationic regions of albumin (Fig. 3b). Future molecular

dynamics simulations of albumin interactions in a dense anionic PEG corona may further elucidate this process.

**MΦ SR-A1 receptors recognize albumin-denaturing but not albumin-stabilizing PS surfaces**. Having characterized the influence of albumin adsorption on PS physicochemical properties and uptake by murine-derived MΦ in vitro, we probed the role of SR-A1 in recognizing PS–BSA complexes. We specifically asked whether class A1 scavenger receptors recognize PS–BSA complexes in a fashion that depends on the folding state of adsorbed albumin. Changes in albumin conformation results in recognition by MΦ SR-A1[21,74], and structural changes decrease the half-life of free albumin in the cardiovascular system[75]. Scavenger receptors contribute to the endocytosis of nucleic acid-functionalized gold NPs[76,77], charged polystyrene NPs[19], and layered silicate platelets[74]. For example, cationic amine (NH₂)-modified polystyrene NPs (58 nm diameter) denatured albumin and became recognized by scavenger receptors expressed on the surface of monkey epithelial cells[19]. In contrast, anionic carboxylate (COOH)-modified polystyrene NPs (60 nm diameter) preserved albumin structure and were not recognized by scavenger receptors in the cited study[19].

While one study demonstrated layered silicate platelets/discs denature albumin and are bound by MΦ scavenger receptors in vitro[74], the relationship between adsorbed albumin folding state and recognition by MΦ SR-A1 has not been examined for soft polymeric nanocarriers. Furthermore, SR-A1 mediated recognition of protein-denaturing soft nanocarrier surfaces has not yet been established as a surveillance mechanism utilized by innate immune cells. Our structural analyses demonstrated that Phos PS preserves the fold of albumin, whereas OH PS partially denatures albumin (Fig. 4). The effect of the MeO PS surface on albumin structure was less clear; however, results from the Trp assay and CD spectroscopy suggested partial unfolding and decreases in secondary structure (Fig. 4). We therefore hypothesized that MΦ SR-A1 recognizes PS surfaces that denature albumin (OH PS and MeO PS) and that albumin-stabilizing surfaces (Phos PS) evade SR-A1 recognition.

To test these hypotheses, we pre-adsorbed albumin to the surface of each of the three PS chemistries and assessed their uptake by MΦ pre-treated with either 1× PBS (negative control) or saturating concentrations of a scavenger receptor competitor, fucoidan (2.5 mg/mL)[19] (Fig. 5a). Fucoidan is a sulfated polysaccharide derived from brown algae that binds to SR-A1 with high specificity[78,79]. SR-A1 (CD204) is highly expressed by RAW 264.7 MΦ (Fig. 5b), and fucoidan has been used as an SR-A1 competitor ligand in this cell type by other efforts seeking to develop peptides with high specificity toward SR-A1[80]. While RAW 264.7 MΦ do not express the entire repertoire of scavenger receptors at high levels[81], its high SR-A1 expression makes it a suitable MΦ cell line for studying the influence of this receptor on nanocarrier uptake. Fucoidan was non-toxic to RAW 264.7 MΦ at the high concentration used to saturate the receptors, as ~85% of the cells remained viable after an 8-h incubation period (Supplementary Fig. 18). Furthermore, these high fucoidan concentrations did not affect the levels of SR-A1 at the cell surface during the time course of our uptake studies (Fig. 5c).

Pre-treating MΦ with fucoidan (Fig. 5a) increased the uptake of pristine PS of each surface chemistry (Fig. 5d). The mechanism by which fucoidan enhances the uptake of pristine polymeric nanocarriers in serum-free conditions is not known, but we suspect that this results from enhanced receptor clustering or related phenomena, due to fucoidan's high molecular weight (MW) and receptor-binding sites in close proximity. Nevertheless, this result, obtained under pristine conditions absent of

protein, holds significant implications on the expected result of an experiment preformed with PS–BSA complexes. In particular, any decreases in PS–BSA uptake mediated by pre-treatment with SR-A1 competitor would suggest that SR-A1 contributes to the non-exclusive uptake of the specified PS–BSA formulation. And this result, if found, would further suggest that SR-A1 particularly recognizes the folding state of adsorbed BSA, since it would demonstrate a decrease in the presence of a reagent that accelerates pristine PS uptake.

To examine whether SR-A1 contributes to the MΦ recognition and uptake of PS–BSA complexes, we administered PS–BSA to MΦ after pre-incubation in serum-free media in the absence or presence of fucoidan (Fig. 5e). The median fluorescence intensity (MFI) of cells treated with OH PS–BSA and MeO PS–BSA decreased after fucoidan pre-treatment (Fig. 5f). This lower MFI was only significantly different than control in the case of OH PS (Fig. 5f). The percentage change in MFI was significantly different when comparing Phos PS–BSA to either OH PS–BSA or MeO PS–BSA (Fig. 5g). In these cases, fucoidan had the opposite effect on uptake of anionic Phos PS versus OH PS–BSA or MeO PS–BSA, which is consistent with a role for SR-A1 in recognizing nanocarriers with surfaces that unfold adsorbed albumin. Decreases in SR-A1-mediated uptake were only observed after saturating SR-A1 with the fucoidan competitor (Supplementary Fig. 19a, b). Therefore, we reason that SR-A1 does not recognize Phos-BSA complexes, and the ability of Phos PS to preserve adsorbed albumin structure is responsible for its evasion of SR-A1 recognition.

The specificity of SR-A1, rather than contributions by other scavenger receptors, is demonstrated by competitive uptake studies performed with an alternative scavenger receptor ligand, polyinosinic acid (Supplementary Fig. 20). Polyinosinic acid is a nucleic acid that competes for binding to SR-A1, gp18 and gp30, as well as other pattern recognition receptors[82]. MΦ pre-treatment with polyinosinic acid decreased OH PS–BSA uptake, but it was not statistically significant (Supplementary Fig. 20a, b). We attribute the weaker effect of polyinosinic acid pre-treatment on OH-PS–BSA uptake to result from its greater number of receptor interactions, which dilute its effect on SR-A1-mediated uptake compared to that of fucoidan at an identical concentration.

Similar results were obtained in a corona exchange experiment (Supplementary Figs. 21 and 22a), where purified PS–BSA complexes were introduced into FBS-supplemented media. Under these conditions, pre-adsorbed BSA (of defined folding state; Fig. 4) can exchange with serum proteins and/or remain adsorbed (Supplementary Fig. 22b). The loss of pre-adsorbed albumin was much greater in the case of the polar surfaces, Phos PS and OH PS, compared to the non-polar MeO PS surface (Supplementary Fig. 22b). In this experiment, pre-treating cells with fucoidan led to a highly significant decrease in the uptake of OH PS–protein complexes (Supplementary Fig. 22c, d). Compared to the denaturing effects of the MeO PS surface, this result suggests that the OH PS denaturing effects are more resistant to protein exchange and/or alter structure of a more heterogeneous population of adsorbed proteins.

We conclude that SR-A1 contributes to the recognition of both OH PS–protein and MeO PS–protein complexes that denature albumin to different extents (Figs. 4 and 5h), whereas Phos PS–protein complexes evade SR-A1 recognition due to its ability to stabilize the structure of albumin (Figs. 4 and 5h).

## Discussion

PS having polar neutral, non-polar neutral, and polar anionic surfaces were found to achieve different organ- and cellular-level biodistributions in vivo. Diverse MΦ populations in the spleen, liver, LNs, lungs, and kidneys were particularly sensitive to PS surface chemistry. Since nanocarrier administration into the

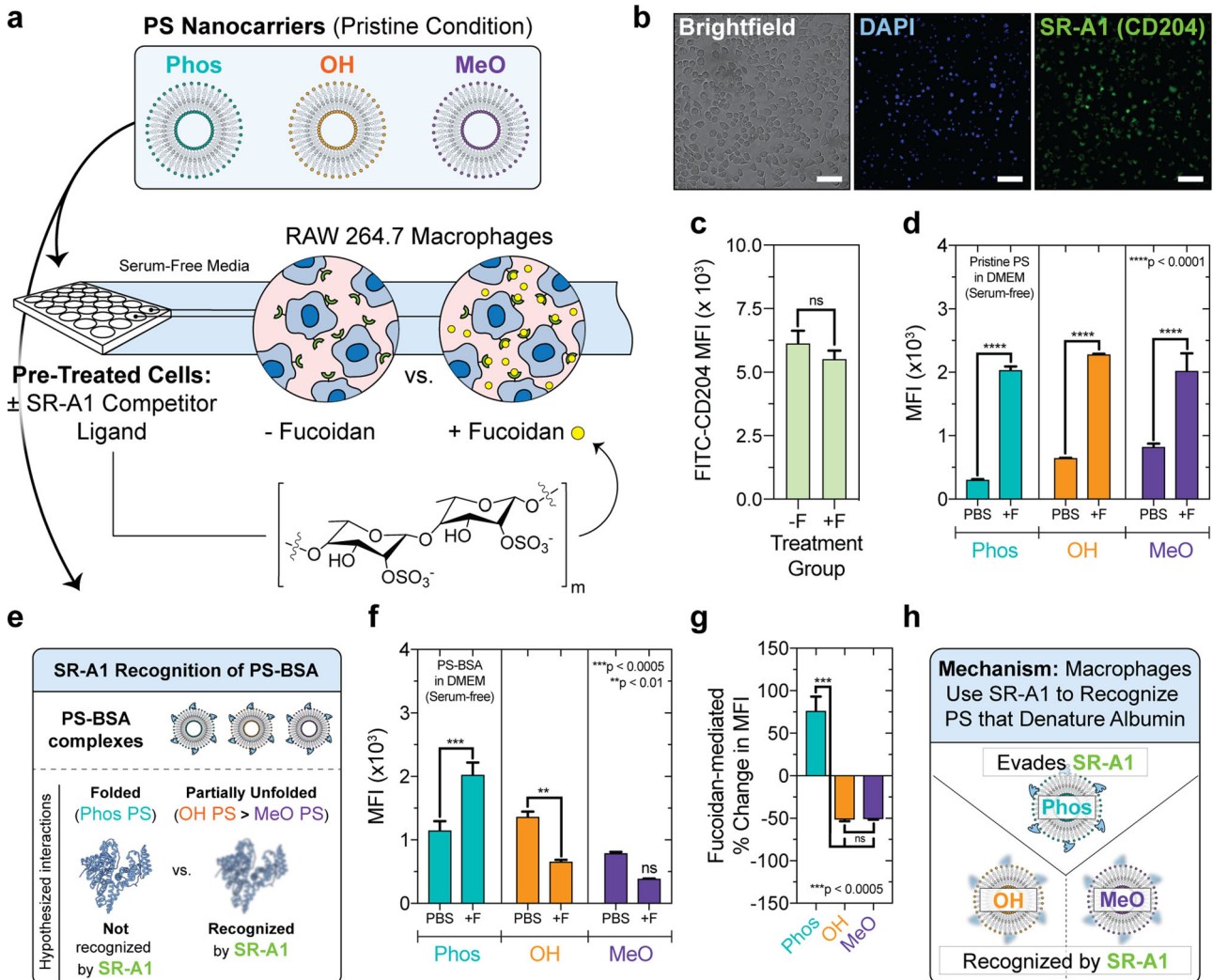

**Fig. 5 SR-A1 recognizes polymersomes with surface chemistry that denatures albumin and contributes to their non-exclusive uptake by macrophages in vitro. a** RAW 264.7 macrophages pre-treated with either Fucoidan (+F) SR-A1 competitor, or PBS, for 30 min at 37 °C. Afterwards, pristine PS (Phos, OH, MeO) were administered for 2 h at 37 °C prior to flow cytometry. **b** Fluorescence microscopy of macrophage SR-A1/CD204 expression. Brightfield, DAPI, FITC (anti-CD204-FITC) channels are shown. Scale bar = 50 µm. **c** Fucoidan (2.5 mg/mL) pre-treatment does not significantly alter surface SR-A1/CD204 expression after 4 h (MFI: median fluorescence intensity; anti-FITC-CD204 fluorescence). Significance was determined by *t* test (unpaired, two-tailed). **d** Fucoidan pre-treatment (2.5 mg/mL) increases pristine PS uptake in vitro. **e**, **f** Differential recognition of PS–BSA complexes by macrophage SR-A1 in vitro. **e** SR-A1 hypothesis illustration. **f** Macrophage uptake of PS–BSA after fucoidan pre-treatment. **g** The percent change in MFI (±fucoidan). Significance was determined by ANOVA using post hoc Tukey's multiple comparisons test. **h** SR-A1 recognition mechanism summary. Serum-free conditions were used for all uptake studies. PS encapsulated hydrophilic $Dex_{70 kDa}$-TMR to quantify vesicle uptake. PS–BSA complexes were formed by incubating PS ([polymer] = 2.5 mg/mL) with 10 mg/mL BSA for 2 h at 37 °C. For **c**, **d**, **f**, **g**, the mean ± s.e.m. is displayed from three biological replicates (*n* = 3). In **d**, **f**, significant differences between ±fucoidan pre-treatment pairs was determined by ANOVA with Sidak's test. All statistical tests used a 5% significance level. ns = not significant, **\*\*p** < 0.01, **\*\*\*p** < 0.0005, **\*\*\*\*p** < 0.0001. Error bars represent s.e.m. (*n* = 3).

protein-rich blood results in the formation of an albumin-rich protein corona, we investigated whether PS surface chemistry differentially modulated the conformation of adsorbed albumin and whether MΦ utilized a highly expressed scavenger receptor to sense these differences in folding state. We hypothesized that MΦ utilize SR-A1 to recognize PS with a partially denatured albumin corona, since unfolded protein is a non-canonical ligand for this receptor.

Through a series of biophysical and biochemical techniques, we demonstrated neutral polar (OH PS) and neutral non-polar (MeO PS) surfaces denatured adsorbed albumin to different extents, whereas an anionic polar (Phos PS) surface preserved albumin structure. While the MeO PS and Phos PS influences on adsorbed protein structure agree with past biophysical protein adsorption

studies using solid core systems, the observation that OH PS denatures albumin represents a significant deviation from certain solid core systems where polar neutral chemical groups stabilize the albumin fold. We attribute this difference to the greater fluidity of the soft neutral PEGylated surface, which promotes the spreading of adsorbed albumin in the absence of significant electrostatic repulsion. SR-A1 competitor studies performed on cultured MΦ confirmed the involvement of SR-A1 as a receptor-mediated mechanism for sensing denaturing PS surfaces (OH PS and MeO PS), whereas the fold-stabilizing anionic PSs (Phos PS) evaded SR-A1 recognition. Put differently, this work established the adsorbed albumin folding state as a molecular determinant of nanocarrier clearance by MΦ scavenger receptors. In the context of our in vivo work, this result suggests that the high MΦ uptake

of the strongly denaturing OH PS is attributable, in part, to SR-A1 detection. This work provides a mechanism that might explain poorly understood changes in nanocarrier biodistributions, for example, the slight changes in lipid/RNA ratios that strongly influence organ accumulation in some nanocarrier platforms[83]. Nevertheless, the mechanism established here can be harnessed for numerous applications. On one hand, controlled albumin denaturation could be used to target the delivery of anti-inflammatory agents or imaging agents to diverse MΦ populations known to overexpress SR-A1, such as the foam cells in atherosclerosis[78]. Alternatively, albumin-stabilizing nanocarriers can be leveraged to evade MΦ recognition to improve their circulation time and their ability to target other cell types. Aside from these observations, we also note that OH PS was taken up at high levels by various immune cell subpopulations in the lungs. This finding was surprising to us, and it may suggest that this nanocarrier type is a useful chassis for targeted drug delivery applications seeking to treat respiratory conditions. However, this particular finding appears to be independent of albumin denaturation, since passive targeting was not observed for the MeO PS that also denatured albumin to a certain extent in the biophysical analyses presented in this work. Collectively, this work offers rational design principles for designing high-performing nanocarriers and offers mechanistic insights into the structural immunology of surveillance systems employed by innate immune cells. The principles established here can be leveraged to tune the uptake of PEGylated soft nanocarriers by SR-A1-expressing cells for a variety of applications in targeted drug delivery and immunomodulation.

## Methods

**Protein samples and chemical reagents**. Fatty acid-free BSA was purchased for use in this study (Fisher Scientific). All chemical reagents were purchased from Sigma-Aldrich unless otherwise indicated.

**Synthesis of PEG-*b*-PPS polymers**. To create PSs having anionic and neutral (including polar versus non-polar neutral surfaces), PEG-*b*-PPS block copolymer (BCP) variants were synthesized to terminate the hydrophilic block with phosphate (Phos, -PO$_4$), hydroxyl (OH, -OH), and methoxy (MeO, -OCH$_3$) chemical groups. To permit the self-assembly of vesicular PS, the hydrophilic mass fraction ($f_{PEG}$) of each polymer is in the range of $0.22 < f_{PEG} < 0.35$. Phos-PEG$_{23}$-*b*-PPS$_{39}$-Bn, HO-PEG$_{23}$-*b*-PPS$_{40}$-Bn, and MeO-PEG$_{17}$-*b*-PPS$_{28}$-Bn BCPs were synthesized and characterized as described below. The benzyl cap on the PPS block is denoted by Bn.

*MeO-PEG$_{17}$-b-PPS-Bn synthesis*: Poly(ethylene glycol) methyl ether (mPEG$_{17}$; 750 g/mol MW) in toluene was dried by azeotropic distillation (20 g, 26.67 mmol, 1 EQ) using a Dean-Stark trap. After water removal, the dried mPEG solution was cooled to room temperature under vacuum. The reaction vessel was subsequently purged with N$_2$. The solution was stirred, triethylamine (18.58 mL, 133.35 mmol, 5 EQ) was added, and was cooled to 0 °C. Methanesulfonyl chloride (10.32 mL, 133.35 mmol, 5 EQ) was diluted into 100 mL toluene and was added dropwise to the cooled, stirred solution. After stirring the solution overnight under N$_2$, the mixture was vacuum filtered over a celite to remove salt, and rotary evaporation was performed subsequently to remove toluene. The resulting product was precipitated in cold diethyl ether, and the precipitate was recovered by filtration, and dried under vacuum after washing with cold diethyl ether. The mesylate-functionalized mPEG$_{17}$ (mPEG$_{17}$-OMs) produced was stored under N$_2$, until it was used to form thioacetate-functionalized mPEG$_{17}$ (mPEG$_{17}$-TAA). The aforementioned mPEG$_{17}$-TAA (5.0 g, 6.67 mmol, 1 EQ) was dissolved in anhydrous dimethylformamide (DMF) within a N$_2$-evacuated round bottom flask, which was then subjected to the sequential addition of potassium carbonate (3.69 g, 26.68 mmol, 4 EQ) and thioacetic acid (1.9 mL, 26.68 mmol, 4 EQ), and the reaction proceeded overnight with stirring. Salt was removed via the aforementioned vacuum filtration procedure. Rotary evaporation was performed to remove DMF, and the product was dissolved in tetrahydrofuran (THF) and was passed through an aluminum oxide column. Rotary evaporation was performed to remove THF and concentrate the product. The concentrated product was precipitated in cold diethyl ether, recovered by filtration, washed with cold diethyl ether, dried under vacuum, and the mPEG$_{17}$-TAA product was stored under N$_2$. Afterwards, mPEG$_{17}$-*b*-PPS was synthesized using the mPEG$_{17}$-TAA macroinitiator. Sodium methoxide (0.5 M prepared in methanol, 0.44 mmol, 1.1 EQ) was added to mPEG$_{17}$-TAA (0.5 g, 0.4 mmol, 1 EQ) to deprotonate mPEG$_{17}$-TAA. This deprotonated mPEG$_{17}$-TAA initiated the living anionic ring-opening polymerization of propylene sulfide (1.0 mL, 12.25 mmol, 30 EQ). Benzyl bromide (1.44 mL, 2.0 mmol, 5 EQ) was added to endcap the terminal thiolate, and the

resulting mixture was stirred overnight. Rotary evaporation was performed to remove DMF. Methanol precipitation was performed to recover the product, which was subsequently dried under vacuum.

*HO-PEG$_{23}$-b-PPS-Bn and Phos-PEG$_{23}$-b-PPS-Bn synthesis*: The preparation of α-tosyl-ω-hydroxyl PEG$_{23}$ is adapted from an established procedure for the synthesis of heterobifunctional PEG[84]. PEG$_{23}$ (25 g, 25 mmol, 1 EQ) was dried in toluene by azeotropic distillation (as described for the synthesis of MeO-PEG$_{17}$-*b*-PPS), the vessel containing dried PEG$_{23}$ was purged with N$_2$, and dissolved in anhydrous dichloromethane (DCM). The DCM-dissolved PEG$_{23}$ was placed on an ice bath afterwards and was subjected to vigorous stirring. Silver (I) oxide (8.69 g, 37.5 mmol, 1.5 EQ), potassium iodide (2.99 g, 18.0 mmol, 0.72 EQ), and *p*-Toluenesulfonyl chloride (5.0 g, 26.25 mmol, 1.05 EQ) were added to the vigorously stirring PEG$_{23}$ (in DCM, on ice). The reaction was allowed to proceed for 2 h. After 2 h, the sample was removed from its ice bath, and the reaction proceeded at room temperature overnight under N$_2$. On the next day, salts and silver (I) oxide was removed from the crude product via vacuum filtration over a celite filter. Afterwards, DCM was removed by rotary evaporation. Cold diethyl ether was added to precipitate the product. The precipitate was recovered by filtration, washed with cold diethyl ether, and dried under vacuum. Tosylate-functionalized PEG (HO-PEG$_{23}$-OTs) results from this procedure and this product was stored under N$_2$. $^1$H-NMR (400 MHz, DMSO): δ 7.78–7.72 (d, 2H), 7.47–7.43 (d, 2H), 4.55–4.49 (t, 1H, OH), 4.09–4.06 (t, 2H, CH$_2$-SO$_2$), 3.49–3.46 (s, 180H, PEG backbone), 2.40–2.38 (s, 3H, CH$_3$).

Benzyl mercaptan (42.0 μL, 0.35 mmol, 1 EQ) was dissolved in DMF. Sodium methoxide (0.5 M in methanol; 0.39 mmol, 1.1 EQ) was subsequently added to deprotonate benzyl mercaptan. This deprotonated benzyl mercaptan was used to initiate the living anionic ring-opening polymerization of propylene sulfide (1.0 mL, 12.25 mmol, 7.5 EQ). HO-PEG$_{23}$-OTs (3.47 g, 2.65 mmol, 7.5 EQ) was added to endcap the terminal thiolate of the polymerized propylene sulfide and was stirred overnight. Rotary evaporation was performed to remove DMF and the product was recovered by methanol precipitation. The recovered precipitate was dried under vacuum. $^1$H-NMR (400 MHz, CDCl$_3$): δ 7.32–7.29 (d, 4H, ArH), 3.64–3.60 (s, 86H, PEG backbone), 2.96–2.81 (m, 80H, CH$_2$), 2.66–2.54 (m, 40H, CH), 1.40–1.33 (d, 119H, CH$_3$).

The synthesis of Phos-PEG$_{23}$-*b*-PPS-Bn proceeded from the aforementioned HO-PEG$_{23}$-*b*-PPS-Bn polymer. Lyophilized compound (1.0 g, 0.29 mmol, 1 EQ) was massed into an evacuated and N$_2$ purged vessel, dissolved in 10 mL of anhydrous THF, and was flushed with N$_2$ for 1 h. After this time period, the dissolved HO-PEG$_{23}$-*b*-PPS-Bn solution was cooled to 0 °C. This cooled HO-PEG$_{23}$-*b*-PPS-Bn solution was vigorously stirred and was subjected to the dropwise addition of phosphorous (V) oxychloride (34.3 μL, 0.37 mmol, 1.27 EQ). After phosphorous (V) oxychloride addition, the solution was removed from the ice bath, and the reaction was allowed to proceed overnight at room temperature under N$_2$. On the following day, 5.0 mL of Milli-Q water was added to quench the reaction and was stirred for 5 min. The product was extracted with DCM, dried over sodium sulfate, filtered, and was then recovered by precipitation. This precipitated product was dried under vacuum overnight. $^1$H-NMR (400 MHz, CDCl$_3$): δ 7.24–7.22 (d, 4H, ArH), 3.58–3.54 (s, 71H, PEG backbone), 2.88–2.73 (m, 79H, CH$_2$), 2.59–2.47 (m, 40H, CH), 1.33–1.26 (d, 117H, CH$_3$).

The resulting polymers were characterized by nuclear magnetic resonance (NMR). Supplementary Table 1 displays a summary of all synthesized polymers used in this study. The chemical structure of each polymer is displayed in Supplementary Fig. 1. Sample H NMR spectra are displayed in Supplementary Fig. 2a–c.

**Nanostructure preparation and purification**. PS nanocarriers were self-assembled from PEG-*b*-PPS BCPs (Supplementary Table 1) using the FNP technique as described previously[27,28]. Briefly, 20 mg of polymer was dissolved in 500 μL of THF (4.0% w/v), loaded into a 1-mL plastic disposable syringe, and was impinged against 500 μL of sterile 1× PBS. PS formulations were subjected to four impingements. The final impingement was performed into a 1.5 mL reservoir of sterile 1× PBS. Organic solvent was removed by desiccation overnight. Nanocarriers were prepared under sterile conditions for all studies performed in vitro and in vivo.

**Nanostructure physicochemical characterization**. All PS formulations were characterized by TEM, DLS, ELS, and small-angle x-ray scattering (SAXS). For DLS, PS samples were briefly vortexed, and a semi-dilute sample was prepared via 1:100 dilution in 1× PBS. DLS size and polydispersity measurements were obtained on a Zetasizer Nano instrument (Malvern Instruments). The number average size and polydispersity index were calculated. For ELS, PS samples were diluted 1:10 in 0.1× PBS. All PBS stocks used for sample dilution were filtered through a 0.2-μm nylon filter. The characterization of PS formulations by DLS, ELS, and SAXS is summarized in Table 1.

**Cryogenic TEM**. Prior to plunge-freezing, 200 mesh Cu grids with a lacey carbon membrane (EMS Cat# LC200-CU-100) were glow discharged in a Pelco easiGlow glow discharger (Ted Pella) using an atmosphere plasma generated at 15 mA for 15 s with a pressure of 0.24 mbar. This treatment created a negative charge on the carbon

membrane, allowing liquid samples to spread evenly over the grid. Four microliters of the sample was pipetted onto the grid and blotted for 5 s with a blot offset of +0.5 mm, followed by immediate plunging into liquid ethane within a FEI Vitrobot Mark III plunge freezing instrument (Thermo Fisher Scientific). Grids were then transferred to liquid nitrogen for storage. The plunge-frozen grids were kept vitreous at –180 °C in a Gatan Cryo Transfer Holder model 626.6 (Gatan) while viewing in a JEOL JEM1230 LaB6 emission TEM (JEOL USA) at 100 keV. Image data were collected by a Gatan Orius SC1000 CCD camera Model 831 (Gatan).

**Nanocarrier morphology conformation by SAXS.** SAXS was performed at the DuPont-Northwestern-Dow Collaborative Access Team (DND-CAT) beamline at the Advanced Photon Source at Argonne National Laboratory (Argonne, IL, USA). SAXS was performed using a sample-to-detector distance of approximately 7.5 m. Samples were subjected to 10 keV ($\lambda = 1.24$ Å) collimated x-rays and were analyzed in the $q$-range (0.001–0.5 Å$^{-1}$). A 3-s exposure time was used to characterize nanocarriers. The momentum transfer vector ($q$) is defined by Eq. 1, where $\theta$ is the scattering angle:

$$q = 4\pi \frac{\sin(\theta)}{\lambda} \quad (1)$$

The $q$-range was calibrated using silver behenate diffraction patterns. PRIMUS 2.8.3. was used for data reduction (solvent/buffer background scattering removal). SasView 5.0 was used for model fitting. The PS nanostructure morphology was confirmed by fitting the scattering profiles to a spherical vesicle model (Eq. 2)[33]. The PS nanostructure morphology was confirmed by fitting the scattering profiles to a spherical vesicle model (Eq. 2)[33].

$$P(q) = \frac{\phi}{V_{shell}} \left[ \frac{3V_c(\rho_{solv} - \rho_{shell})j_1(qr_c)}{qr_c} + \frac{3V_t(\rho_{shell} - \rho_{solv})j_i(qr_t)}{qr_t} \right]^2 + b \quad (2)$$

where $\phi$ is the shell volume fraction (dimensionless), $V_{shell}$ is the shell volume, $V_c$ is the core volume, $V_t$ is the total volume, $\rho_{solv}$ is the solvent scattering length density (10$^{-6}$ Å$^{-2}$), $\rho_{shell}$ is the shell scattering length density (10$^{-6}$ Å$^{-2}$), $r_c$ is the core radius (Å), $r_t$ is the total radius (Å), $b$ is the background (cm$^{-1}$), and $j_1$ is the spherical Bessel function described by Eq. 3:

$$j_1 = \frac{(\sin(x) - x\cos(x))}{x^2} \quad (3)$$

The shell thickness ($r_{st}$) is described by Eq. 4:

$$r_{st} = r_t - r_c \quad (4)$$

$P(q)$ is scaled to units of cm$^{-1}$ sr$^{-1}$. Diameter measurements by DLS were used to guide the selection of initial parameter values. The Levenberg–Marquardt algorithm was used to fit parameter values (chi square ($X^2$) minimization). A $X^2$ value of <1 indicates a good model fit. Kratke and Guinier analyses were performed to analyze albumin conformation upon adsorption to PS nanocarriers.

**Rheological characterization of PEG-b-PPS PSs.** An Oscillatory HR-3 rheometer (TA Instruments) equipped with a cone-and-plate geometry (4° angle, 25 mm diameter, and Gap, 110 μm) was utilized for rheology measurements. MeO PSs were used as a model PEG-b-PPS PS in these studies. All of the samples (0.3 mL of 20 mg/mL MeO PS suspensions) were placed on the Peltier plate at 25 °C, and a solvent trap was used to delay the evaporation of the sample. Initially, the linear viscoelastic region of MeO PS suspensions was determined using strain sweep measurements (0.1–100%) at 25 °C (keeping the angular frequency constant at 1 rad/s). Afterwards, the angular frequency sweep measurements were performed within the viscoelastic region. The frequency dependence of the dynamic moduli $G'$ and $G''$ were measured at 25 °C with angular frequency sweep measurements from 0.1 to 50 rad/s and 10% oscillatory strain.

**Loading of hydrophilic cargo.** In all, 70 kDa Dex-TMR (Life Technologies) were prepared in sterile 1× PBS at stock concentration of 6.25 mg/mL. For hydrophilic loading studies, 375 μg of stock Dex$_{70 kDa}$-TMR was diluted in sterile 1× PBS to a final volume of 500 μL and was impinged against 20 mg of the specified PEG-b-PPS polymer dissolved in 500 μL THF following FNP procedures described above. Dex-TMR-loaded PSs were purified by size exclusion chromatography using a Sepharose 6B column. Opaque, polymer-containing fractions were pooled and spin concentrated to the input sample volume. The Dex-TMR concentration in the specified PS sample was determined from a linear regression model fit using fluorescence measurements ($\lambda_{Ex} = 555$ nm, $\lambda_{Em} = 580$ nm) against a Dex-TMR calibration curve. Dex-TMR concentrations of 0, 10, 25, 50, 75, 100, 150, and 200 μg/mL prepared in 1× PBS was used for calibration. Fluorescence measurements were recorded using a SpectraMax M3 microplate reader (Molecular Devices). Dex-TMR encapsulation efficiency (EE), provided as a percentage, was calculated using Eq. 5, where $M_i$ is the input mass of the cargo and $M_u$ is the mass of the unencapsulated cargo.

$$EE = \frac{(M_i - M_u)}{M_i} * 100 \quad (5)$$

**Protein structure representation and graphics.** The Pfam[85–89] domains were represented with respect to their positions in the crystal structure solved for BSA

(PDB ID: 4F5S[52]). UCSF Chimera[90] software (version 1.12) was used to prepare protein structure graphics. For the electrostatic depiction of BSA, Coulombic surface coloring was displayed after computing the electrostatic potential in accordance with Coulomb's law.

**Bioinformatic analysis.** Phyre2[91] software was used to predict the three-dimensional structure of mouse serum albumin. Mouse serum albumin (UniProt accession: P07724) was used as the input primary structure (amino acid sequence) for this three-dimensional structural prediction. A custom python script (written in python version 2.7.12) was used to perform all pairwise structural alignments. This script uses UCSF Chimera structural alignment tool for pairwise structural superposition and RMSD calculations on the backend[92]. The solved crystal structures for BSA (PDB ID: 4F5S[52]) and human serum albumin (PDB ID: 1E78[51]) were used in protein structural alignments.

**Protein corona formation, isolation of PS–protein complexes, and determination of adsorbed protein concentration.** Unless specified otherwise, PSs (2.5 mg/mL polymer concentration) were incubated with 0.1, 1.0, or 10.0 mg/mL BSA, or with complete Dulbecco's Modified Eagle Medium (CDMEM) supplemented with 10% FBS, at 37 °C, 80 rpm for 2 h. PS–protein complexes were isolated by ultracentrifugation at 25,000 × g, 4 °C for 30 min in a Beckman Coulter Optima MAX-XP ultracentrifuge operating under vacuum. The supernatant was removed, and PS–protein complexes were gently washed with sterile 1× PBS. This washing procedure was repeated twice to remove free protein. In the final step, PS–BSA complexes were gently resuspended in 1× PBS. The concentration of adsorbed proteins was determined using the Pierce 660 nm assay (Thermo Scientific) calibrated against a BSA concentration series (0, 0.125, 0.250, 0.5, 0.75, 1.0, 1.5, and 2.0 mg/mL). Absorbance of 660 nm light was measured using a SpectraMax M3 microplate reader (Molecular Devices).

**Analysis of adsorbed serum protein composition.** Proteins were prepared in Laemmli[93] buffer containing 10% β-mercaptoethanol and were heated at 95 °C for 5 min prior to loading in the gel. Protein samples were loaded into 4–20% tris-glycine polyacrylamide gels (Mini-PROTEAN TGX gels, Bio-Rad Laboratories). A 1:5 dilution of the PageRuler Plus pre-stained protein ladder was used as a MW standard (ThermoFisher Scientific). Proteins were separated by SDS-PAGE in 1× SDS running buffer (25 mM Tris, 192 mM glycine, 0.1% SDS, pH 8.3) at 120 V for 70 min. Gels were washed, fixed overnight in a 6:3:1 water:ethanol:acetic acid solution in glass trays, and were subsequently silver stained to non-specifically detect proteins. Silver staining was performed using the Pierce Silver Stain Kit for MS (ThermoFisher Scientific). Gels were imaged on an Epson V39 scanner (Seiko Epson). Grayscale TIFF images (800 dpi) were obtained for each gel.

**Determination of adsorbed protein exchange.** PS nanocarriers (2.5 mg/mL polymer concentration) were incubated with fluorescein isothiocyanate (FITC)-BSA (10 mg/mL) at 37 °C, 80 rpm for 2 h. PS–protein complexes were isolated using the ultracentrifugation procedure described above. The concentration of adsorbed FITC-BSA was determined by fluorescence ($\lambda_{Ex} = 495$ nm; $\lambda_{Em} = 519$ nm), calibrated against a FITC-BSA concentration series. PS-associated protein concentration was determined by subtracting the protein concentration of the supernatant (isolated after the first ultracentrifugation step) from the known input protein concentration during incubation (10 mg/mL). To quantify the exchange of pre-adsorbed FITC-BSA with unlabeled serum proteins during the cell culture experiments performed in serum-supplemented media, the pre-adsorbed samples were prepared in parallel and were processed with the following procedural changes. After isolating and purifying PS–(FITC-BSA) complexes, these samples were gently resuspended in complete DMEM (CDMEM) supplemented with 10% FBS and were incubated at 37 °C, 80 rpm for a second 2 h period. PS–protein complexes were isolated and purified by iterative rounds of ultracentrifugation. The concentration of FITC-BSA in the supernatant (prior to washing steps) was again used to infer FITC-BSA retention by PS in the pellet. Since this supernatant has a CDMEM background (rather than PBS), the FITC-BSA calibration curve was prepared in the original stock of CDMEM supplemented with 10% FBS to account for background differences. The percent decrease in pre-adsorbed FITC-BSA was calculated.

**Tryptophan quenching assay.** Phos PS, OH PS, and MeO PS ([polymer] = 0.25 mg/mL) was incubated with 1 mg/mL BSA at 37 °C, 80 rpm, for 2 h. BSA prepared with PBS or with 5 M guanidinium chloride (Gdn-HCl) were included as controls for folded BSA and chemically denatured BSA, respectively. Gdn-HCl disrupts protein secondary structure[67–69,94–96]. Tryptophan fluorescence ($\lambda_{Ex} = 280$ nm; $\lambda_{Em} = 300$–400 nm) was recorded using a SpectraMax M3 microplate reader (Molecular Devices). To assess the fluorescence of the protein analyte and minimize background contributions from the specific combinations of buffer and/or polymer, the fluorescence of buffer (PBS or Gdn-HCl) supplemented with non-protein components (i.e., PS without protein) was background subtracted from each specific sample. Measurements were obtained in triplicate.

**Limited proteolysis**. BSA (1 mg/mL) prepared in 0.1× PBS was incubated with Phos PS, OH PS, and MeO PS (0.25 mg/mL) for 2 h at 37 °C, 80 rpm. Low salt buffer was used to minimize its influence on ionization required for analysis by MS. BSA prepared in PBS without PSs was used as a control. After pre-incubation, Trypsin gold (100 nM; Promega) was added to each sample and proteolysis proceeded for 2 h at 37 °C, 80 rpm. At each of the six timepoints (0, 10, 20, 30, 60, and 120 min), a 50-μL aliquot was removed and the reaction was quenched with 1% formic acid. The $t_0$ timepoint was prepared prior to protease addition. All samples were stored at −80 °C prior to further analysis. Sinapic acid matrix (purity ≥99.0%; Sigma) was prepared by 70:30 water/acetonitrile with 0.2% trifloroacetic acid. Prior to MS, samples were thawed and prepared 1:1 in sinapic acid matrix. The prepared samples were spotted onto a 384-spot stainless steel target (Bruker) and were dried with a heat gun. MALDI-TOF MS was performed using a Bruker rapifleX™ MALDI Tissuetyper. The baseline subtracted peaks were integrated and the area under the curve (AUC) was normalized to the AUC at $t_0$ (prior to proteolysis). A one-phase decay model (Eq. 6) was fit to each time course to determine the apparent rate of proteolysis ($x$ = time (min); $y_0$ = AUC at $t_0$; $y_p$ = plateau AUC; $k$ = rate of proteolysis (min$^{-1}$)). Rate differences between the free BSA control and BSA adsorbed to Phos PS, OH PS, and MeO PS were assessed.

$$y = (y_0 - y_p)e^{-kx} \qquad (6)$$

**CD spectroscopy of albumin secondary structure**. PSs (50 μg/mL polymer concentration) were incubated with BSA (200 μg/mL) at 37 °C for 2 h prior to performing CD spectroscopy. This BSA/PS ratio permitted the albumin structure to be analyzed with a high signal-to-noise ratio and without interference from PS background. Samples were loaded into a quartz cuvette (0.1 cm path length) at the time of measurement. CD spectroscopy was performed using a Jasco J-815 CD Spectrometer (Jasco). Data from three replicates were collected using continuous scans in the wavelength range of 190–260 nm, with a 2 s digital integration time, 2 nm bandwidth, 0.5 nm data pitch, and 100 nm/min scanning speed. Baseline corrections were made using all buffer and sample components were prepared in the absence of the protein analyte. The chamber temperature was held constant at 37 °C. For all measurements, the high tension voltage was monitored to ensure spectra were recorded in the linear range. Natively folded BSA and heat-denatured BSA (90 °C, 10 min) were included as controls. The mean residue ellipticity (MRE) was calculated using Eq. 7, where $W_r$ is the mean residue weight (calculated from the MW of the protein and the number of amino acid residues), $\theta_{obs}$ is the observed ellipticity, $d$ is the path length of the cuvette (in cm), and $c$ is the protein concentration in g/mL. Our calculations used a primary structure length of 583 residues and a MW of 66,430.3 Da for BSA[50]. The units for MRE are deg cm$^2$/dmol.

$$MRE = \frac{W_r \theta_{obs}}{10dc} \qquad (7)$$

The change in the MRE (ΔMRE) was calculated for PS-adsorbed, and heat-denatured BSA, by subtracting MRE of free BSA control from the MRE determined for these samples. Secondary structure content was determined by analyzing the far ultraviolet region (<250 nm) of the obtained spectra. The molar ellipticity at 208 nm ($\theta_{208}$)[97,98] was used to calculate the α-helix content using Eq. 8[99]:

$$\% \ \alpha \ \text{helicity} = \frac{-[\theta]_{208} - 4000}{33,000 - 4000} * 100 \qquad (8)$$

**RAW 264.7 MΦ cell culture**. Murine RAW 264.7 MΦ were purchased from ATCC and were shipped frozen on dry ice. Cells were cultured in T75 polystyrene tissue culture flasks (BD Biosciences) in CDMEM at 37 °C, 5.0% CO$_2$. CDMEM was prepared by supplementing DMEM (Life Technologies) with 10% FBS (Gibco) and 1% penicillin/streptomycin antibiotics (Life Technologies). Mechanical scraping was used to passage cells after the cells reached 70–80% confluency. Prior to mechanical detachment, the cell culture media was aspirated, and cells were washed twice with 1× PBS.

**Cell viability assay**. Cell viability was assessed by measuring mitochondrial activity using the MTT (3-(4,5-dimethylthiazolyl-2)-2,5-diphenyltetrazolium bromide) assay[100]. This assay quantifies the conversion of the MTT reagent into formazan crystals, which is a reaction carried out by NAD(P)H-dependent oxidoreductases. Briefly, 25,000 cells/well were seeded in a tissue culture-treated 96-well plate and were allowed to adhere overnight at 37 °C, 5.0% CO$_2$ in CDMEM. The following day the media was replaced with either CDMEM or DMEM lacking FBS (as specified), supplemented with either 2.5 mg/mL fucoidan or the specified PS formulation at 0.01, 0.1, or 1.0 mg/mL. PBS-treated cells were used as negative controls for toxicity (i.e., healthy cell controls). Cells were incubated under the specified conditions at 37 °C, 5.0% CO$_2$ for the specified duration of time. At that time, cells were treated with thiazolyl blue tetrazolium bromide (MTT) (Sigma) at a working concentration of 0.5 mg/mL reagent prepared in 1× PBS. Cell viability was assessed after a 5 h incubation period with the reagent. The absorbance of 570 nm light was measured in five replicates. Percent viability was calculated by dividing the absorbance of the experimental sample by the mean fluorescence of the PBS treatment group.

**Assessment of scavenger receptor A1 expression in vitro**. RAW 264.7 MΦ were seeded at appropriate density for analysis by fluorescence microscopy or flow cytometry (defined elsewhere in this "Methods" section). An FITC-conjugated rat anti-mouse CD204 monoclonal antibody (Bio-Rad) was prepared 1:200 in DMEM or CDMEM. MΦ were treated with antibody for 30 min at 37 °C, 5.0% CO$_2$. After incubation, the antibody-containing media was aspirated and cells were washed twice with 1× PBS. Cells were prepared for either fluorescence microscopy or flow cytometry as specified elsewhere in this "Methods" section.

**Scavenger receptor competition assays**. RAW 264.7 MΦ were adhered to tissue culture-treated 24-well polystyrene plates (Falcon) at a density of 100,000 cells/well and were incubated overnight at 37 °C, 5.0% CO$_2$ prior to performing uptake studies. Thirty minutes prior to the start of the uptake study (nanocarrier administration), media was aspirated, the cells were washed twice with 1× PBS, and were placed in fresh cell culture media. For studies performed under serum-free conditions, CDMEM was replaced with DMEM lacking 10% FBS but containing 1% penicillin/streptomycin antibiotics. For competitive adsorption studies, fresh CDMEM was used. Where cell pre-treatment with scavenger receptor competitors is indicated, the 30 min pre-incubation was performed with the specified media type (CDMEM or DMEM), prepared with the specified concentration (0.025, 0.25, or 2.5 mg/mL) of fucoidan (Cayman Chemical), or polyinosinic acid potassium salt homopolymer (Sigma-Aldrich). Dex-TMR PS (of the specified surface chemistry; [polymer] = 2 mg/mL) were incubated with either 1× PBS (referred to here as the pristine condition) or BSA (0.1, 1.0, or 10.0 mg/mL) at 37 °C, 80 rpm for 2 h prior to beginning the uptake study. Cells were treated with the specified PS formulation ([polymer]$_{well}$ = 0.5 mg/mL) and were left to incubate at 37 °C for 2 h. At that time, the media was aspirated, cells were washed twice in 1× PBS, and were harvested by mechanical scraping. Cells were stained with Zombie Aqua fixable viability dye (Biolegend) for 15 min at 4 °C. The cells were subsequently washed and stained with intracellular fixation buffer (Biosciences) prior to uptake assessment by flow cytometry.

**Confocal microscopy**. Cells were cultured in complete DMEM in 8-well chamber slides at a density of 30,000 cells/well (400 μL well volume). Cells were allowed to adhere overnight at 37 °C. Prior to imaging, cells were washed twice in 1× PBS and placed into a final 40 μL of 1× PBS. The nucleus of live cells were stained using NucBlue (one drop/well). For studies assessing lysosomal trafficking, cells were also stained with LysoTracker green dye (Invitrogen). Confocal microscopic images of cultured cells were obtained using a Leica TCS SP8 confocal microscope (×63 objective magnification). Brightfield and cell nuclei images were obtained. Cell nuclei were imaged using a diode laser ($\lambda_{Ex}$ = 405 nm, $\lambda_{Em}$ = 460 nm).

**Live cell high-throughput fluorescence microscopy**. Cells were seeded at 20,000 cells/well in SensoPlate™ 96-well glass bottom microplates (Grenier Bio-One) and were allowed to adhere overnight at 37 °C, 5.0% CO$_2$ prior to imaging. Cells were washed with PBS and were treated with anti-CD16/CD32 to block FcRs. Cells were subsequently washed with PBS and were stained with FITC-conjugated anti-CD204 (SR-A1) to assess SR-A1 expression. After antibody staining, cells were briefly washed and fixed. Wide-field fluorescence microscopic images were obtained at ×40 magnification in high-throughput using an ImageXpress High Content Imaging Robotic Platform (Molecular Devices) integrated with a CRS CatX robot and Cytomat CO$_2$ incubator. A z-stack of images was obtained from four sites in each well. Images were obtained with transmitted light (50%) and fluorescence filters (DAPI, FITC, CY5, and Texas Red).

**Mouse models**. C57BL/6J mice were purchased from The Jackson Laboratory and were fed a standard diet. Mice were sheltered and maintained in the Center for Comparative Medicine at Northwestern University at 18–23 °C with 40–60% humidity and 12 h/12 h dark/light cycle. All experiments conducted in vivo were performed in accordance with animal protocols approved by the Institutional Animal Care and Use Committee (IACUC) at Northwestern University. Female mice (90-day old) were allocated randomly to the experimental and/or control groups.

**Polymersome administration to C57BL/6J mice and imaging in vivo**. Phos, OH, and MeO PSs encapsulating DiR were prepared by FNP under sterile conditions. Unencapsulated free dye was removed using a sterile LH-20 column (Sigma). In all, 1× PBS (control) or DiR PS (Phos, OH, or MeO; 6.75 mg/mL polymer concentration) were administered IV via retro-orbital injections. Each group contained five mice, and each mouse received a 100-μL injection. Mice were euthanized 4 h after administration. Whole blood was collected in heparin-treated tubes by cardiac puncture. The spleen (SP), liver (LV), LNs, kidneys (K), lungs (LG), and heart (H) was dissected after performing a whole-body PBS perfusion. A total of six LNs were collected per mouse (2 axial, 2 brachial, and 2 inguinal). SP, LV, K, LG, and H organs were imaged using the IVIS in vivo imaging system (PerkinElmer). Organs dissected from a single animal were imaged together on a petri dish. DiR PS were detected using $\lambda_{Ex}$ = 745 nm, $\lambda_{Em}$ of 800 nm, and a 0.5-s exposure time. IVIS data were processed using the Living Image software version 4.5.5. (PerkinElmer). The radiant efficiency ((p/s/cm$^2$/sr)/(μW/cm$^2$)) was determined in a circular region of interest (ROI). A 1.9-cm diameter ROI was used for

spleen, kidney, lung, and heart measurements, whereas liver radiant efficiency measurements were made in a 3.0-cm diameter ROI. Background measurements were obtained from an ROI position in an empty region of the petri dish in each image. Radiant efficiency measurements were adjusted by subtracting the average radiant efficiency (($p/s/cm^2/sr$)/($\mu W/cm^2$)) of the background from the total radiant efficiency (($p/s$)/($\mu W/cm^2$)) of the organ and multiplying this value by the product of the ROI area ($cm^2$) and a factor computed as (total radiant efficiency)/ ((average radiant efficiency) × (ROI area)).

Single-cell suspensions were prepared from dissected spleen, liver, LNs, kidneys, and lungs following previously established protocols in our laboratory. Single-cell suspensions were then prepared for flow cytometric analysis.

**Flow cytometric analysis**. For all cellular staining with antibodies, cells were washed with PBS. FcR binding was subsequently blocked by incubating cells with anti-mouse CD16/32 (Biolegend) and cells were simultaneously stained with fixable Zombie Aqua (used for in vitro experiments) or fixable Zombie Violet (used for in vivo experiments) viability dye (Biolegend) for 20 min at 4 °C to distinguish live cells from dead cells. For SR-A1 expression tests in vitro, cells were washed in PBS and were treated with anti-mouse SR-A1 (Bio-Rad Laboratories). For flow cytometric analysis of single-cell suspensions prepared from harvested organs (mouse studies), cells were stained using cocktails of fluorophore-conjugated antibodies.

The antibody cocktails are as follows. Spleen and LNs: BUV395 anti-CD45, BV510 anti-CD3 anti-CD19 anti-NK-1.1, PerCP/Cy5.5 anti-CD11b, PE anti-CD11c, PE/Dazzle 594 anti-F4/80, APC anti-CD169, and AlexaFluor 700 anti-Ly-6C. Liver and kidneys: BUV395 anti-CD45, BV510 anti-CD3 anti-CD19 anti-NK-1.1, PerCP/Cy5.5 anti-CD11b, PE anti-CD11c, PE/Dazzle 594 anti-F4/80, APC anti-IA/IE, and AlexaFluor 700 anti-Ly-6G. Lungs: BUV395 anti-CD45, BV510 anti-CD24, PerCP/Cy5.5 anti-CD11b, PE anti-CD11c, PE/Dazzle 594 anti-CD64, APC anti-IA/IE, and AlexaFluor 700 anti-Ly-6C. DiR-loaded PS of each surface chemistry were used to trace the cellular biodistribution of the nanocarriers in our in vivo studies. For each antibody panel, both single color fluorescence compensation controls and fluorescence minus one (FMO) controls were prepared. Antibodies were diluted as recommended by the manufacturer.

Flow cytometry was performed using the BD LSRFortessa 6-Laser Flow Cytometer (16 color compatible instrument). For in vitro studies, 10,000 single, live cell events were recorded. For in vivo studies, a minimum of 750,000 single, live cell events were recorded (with the exception of the liver, for which a minimum of 250,000 events were recorded). The resulting data were analyzed using the Cytobank analysis suite[101]. For all processing of flow cytometric data using Cytobank software in this manuscript, Cytobank Community was used unless otherwise indicated. The population of live, single cells were gated using SSC, FSC, and the viability staining. For mouse cellular biodistribution studies, the FMO controls were used to objectively gate on specific cellular populations (Supplementary Tables 4 and 5). The percentage of NP-positive cells or SR-A1+ cells was determined using a false positive rate of 1.5% (in vitro) or 2% (in vivo studies). This false positive rate was set by gating PBS-treated cells. The MFI was normalized by subtracting the background MFI determined in untreated cells. Therefore, the reported MFI is computed from the fluorescence above background that results from the cellular uptake of dye-loaded nanocarriers.

**Statistics and reproducibility**. Prism software (version 8.4.1; GraphPad Software) was used to perform all statistical analyses. The statistical tests used to conduct each analysis in the study are described in the corresponding figure legends. A 5% significance level was used for all statistical tests. For all micrographs displayed (Cryo-TEM, confocal microscopy, and fluorescence microscopy) and SDS-PAGE gels, representative images are displayed from a minimum of two independent experiments/analyses.

**Reporting summary**. Further information on research design is available in the Nature Research Reporting Summary linked to this article.

## Data availability
Relevant data will be made available by the authors upon request.

## Code availability
The code used to perform protein structural alignments in high throughput is available upon request.

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

## Acknowledgements

M.P.V. gratefully acknowledges support from the Ryan Fellowship and the International Institute for Nanotechnology at Northwestern University. We thank Dr. Sean Allen (California NanoSystems Institute, University of California, Los Angeles) for his advice regarding antibody panels for the in vivo flow cytometric analysis of immune cells. We thank Eric W. Roth (Northwestern University) for his assistance and expertise with Cryo-TEM. This research was supported by the National Science Foundation CAREER Award no. 1453576 and the National Institutes of Health Director's New Innovator Award no. 1DP2HL132390-01. This research used resources of the Advanced Photon Source, a U.S. Department of Energy (DOE) Office of Science User Facility operated for the DOE Office of Science by Argonne National Laboratory under Contract No. DE-AC02-06CH11357. SAXS experiments were performed at the DuPont-Northwestern-Dow Collaborative Access Team (DND-CAT) located at Sector 5 of the Advanced Photon Source (APS). DND-CAT is supported by Northwestern University, E.I. DuPont de Nemours & Co., and The Dow Chemical Company. This work made use of the BioCryo facility of Northwestern University's NUANCE Center, which has received support from the Soft and Hybrid Nanotechnology Experimental (SHyNE) Resource (NSF ECCS-1542205); the MRSEC program (NSF DMR-1720139) at the Materials Research Center; the International Institute for Nanotechnology (IIN); and the State of Illinois, through the IIN. It also made use of the CryoCluster equipment, which has received support from the MRI program (NSF DMR-1229693). We thank the Robert H. Lurie Comprehensive Cancer Center (RHLCCC) of Northwestern University in Chicago, IL for the use of the Keck Biophysics resources. The Lurie Cancer Center is supported in part by an NCI Cancer Center Support Grant #P30 CA060553. This work was further supported by the Northwestern University RHLCCC Flow Cytometry Facility and a Cancer Center Support Grant (NCI CA060553). Microscopy was performed at the Biological Imaging Facility at Northwestern University, graciously supported by the Chemistry for Life Processes Institute, the NU Office for Research, and the Rice Foundation. This work used resources of the Northwestern University Structural Biology Facility, which is generously supported by NCI CCSG P30 CA060553 awarded to the Robert H. Lurie Comprehensive Cancer Center. Imaging work was performed at the Northwestern University Center for Advanced Molecular Imaging generously supported by NCI CCSG P30 CA060553 awarded to the Robert H. Lurie Comprehensive Cancer Center.

## Author contributions

M.P.V. and E.A.S. contributed to the conception and design of the study; statistical analysis; and writing the manuscript, figure creation, and the graphical presentation of the data. M.P.V. designed and conducted all experiments; analyzed the data obtained from each experiment; prepared polymersome formulations; characterized the physico-chemical properties of all polymersome formulations; performed small angle x-ray scattering (SAXS) nanocarrier characterization and protein adsorption experiments at Argonne National Laboratory; analyzed and modeled SAXS data; performed and analyzed IVIS; designed antibody panels for cellular biodistribution analysis of cells derived from mouse spleen, lymph node, liver, kidney, and lung tissue; performed and analyzed all protein adsorption studies and biophysical characterization of protein structure; performed bioinformatic analyses; and performed and analyzed flow cytometric, high-throughput microscopic, and toxicity experiments. N.B.K. and M.F. synthesized PEG-*b*-PPS polymers. M.P.V., Y.L., S.B., and T.S. performed mouse studies and collected and processed mouse organs. Y.L. performed retro-orbital intravenous injections and whole-body perfusions. M.P.V. and J.O.N. completed nanocarrier uptake experiments in vitro. S.B. performed confocal microscopy and rheology experiments.

## Competing interests

The authors declare no competing interests.
