## [Peer Review File · Nature Communications]

REVIEWER COMMENTS

Reviewer #1 (Remarks to the Author):

The manuscript describes the effects of modification of nanoparticles surface on their clearance by macrophage subsets. Overall, it is well written, logical and easy to read. The experimental procedures are thorough and well described. To achieve their aims, the authors synthesised 3 'soft' nanoparticles comprised of functionalized PEG hydrophilic blocks. These particles varied in surface charge, although the OH- and MeO- modified particles were very similar.

The findings are robust, and represent a contribution to the field. Some of the results confirm previous reports, which could be better referenced. A major weakness of the study is the number of nanoparticles examined. It is too small to draw any generalisation, especially with regard to both soft and hard PEGylated nanoparticles synthesised from other materials. Some of the experimental conditions are not optimum (see below). The authors should be more inclusive with their referencing, especially for work that previously reported similar findings, albeit with different nanoparticles.

The following comments are provided for consideration by the authors.

1. Introduction:

a. While albumin is often the most prominent protein in coronas, it is not necessarily the most important. Minor proteins and non-protein molecules can be important determinants of NP fate in vivo. This should at least be acknowledged by the authors.

b. I am unaware of anyone in the field who has championed the concept that albumin is responsible for the stealth effect of PEGylated NP. This statement should be referenced or removed.

c. The author should distinguish between unfolded and denatured albumin. These are different processes with different outcomes. One is reversible while the second is not. Consequently, the interaction with receptors may be very different for these different states of the same protein.

d. Reference 15 does not mention class AI scavenger receptors. Similarly, reference 14 does not include experiments that distinguish the different subclasses of scavenger receptors. The author should carefully check the appropriateness of the references throughout the introduction.

e. The authors state that the role of scavenger receptors in recognising and clearing soft nanocarriers has not been examined to date. That is not correct, unless I misunderstand the meaning of a soft nanocarrier. Perhaps the authors could better define this for the reader.

2. Methodology:

a. The poly dispersion index for each of nanoparticles is reasonable, but this does not align well with the average diameters shown in supplementary figure 4 where the standard deviation for the diameter is 30 to 50% of the mean. Does the supplementary figure show between batch variation or within batch variation? If it is between batch variation, then how was this accommodated between experiments?

b. What effect did the dextran loading have on size/charge?

c. Protein coronas were formed in 10% FBS. There is ample evidence in the literature to show that the corona formed in 10% serum is very different to that formed in 100% serum. Why choose this concentration in the first place? How might it reflect the expected corona following administration of the NP to mice?

3. Results:

a. Supplementary figure 6 shows that 1 mg per ml nanoparticles causes a significant decrease in viability. This is contrary to what is stated in the text.

b. While proteins bind rapidly to most NP, the protein corona changes with time and protein concentration. Those proteins with high concentrations bind first but they are displaced over time by minor components that are lower in concentration but have a greater affinity for the NP. It is during this evolutionary process that most tightly bound proteins unfold as they spread across the surface of the NP. This raises the important question of whether the experiments were performed at equilibrium. Why was a 2 hr incubation chosen?

c. Supplementary figure 7 clearly shows that the stability of the particles were not the same. However this is not what is stated in the text.

d. In figure 5C, there was a significant decrease in cell viability. It is unclear why the authors have chosen 75% viability as a cut-off. This appears random.

- e. In figure 5e, flow cytometry does not distinguish internalised versus surface bound NP. This needs to be considered in both the interpretation of the results and their discussion.
 - f. Use chemical inhibitors to identify specific receptors involved in ligand uptake is problematic. None of these is a highly specific, and all of them have the potential to directly interact with the particle as opposed to the target receptor. The use of cells that do not express target receptors or where the target receptors have been knocked down with siRNA is a much better protocol for identifying specific receptors involved in uptake.
 - g. It should be acknowledged that RAW cells do not express the entire repertoire of scavenger receptors. Consequently, these cells can only be used to identify those receptors that they express. This should be mentioned somewhere in the manuscript.
4. Discussion:
- a. A series of three nanoparticles is not very robust and care should be taken in over extrapolating the results from such a small sample size.
 - b. There is little doubt that SR-A1 receptors are responsible for the uptake of most NP, especially following serum protein binding. This is not new. The author should place their finding in context with the many other papers that have reported the same observation.

Reviewer #2 (Remarks to the Author):

In recent years it has become clear that one of the main challenges of nanomedicines is avoiding rapid clearance to the liver and spleen. Instead of reaching their target, the vast majority of nanomedicines are rapidly cleared from circulation making them completely ineffective. This process is mediated by a "corona" of proteins that adsorb on the surface of nanoparticles in the bloodstream. Vincent and co-workers address this challenge with a comprehensive study nanoparticle surface functionalization, corona formation, macrophage uptake, and in vivo targeting. The manuscript stands out for two reasons: 1) a complete story from the chemistry of functionalization to in vivo experiments and 2) a detailed study of the structure of the adsorbed protein and how changes in structure lead to the observed biodistribution. Of specific importance to the nanomedicine community is the role of PEG, a standard functionalization to avoid clearance, in this process. Overall, it is a careful and important study that should be published.

There were two organizational aspects that I did find detracted from the results.

1) The organization of the paper seems chronological rather than results-directed. For example, the results move from functionalization to biodistribution to macrophage uptake to protein structure. I expect this is the order that experiments were carried out. I can imagine that the observed biodistribution led the team to investigate further ultimately examining protein structure. However I think there would be value in rearranging to order in terms of the complexity of the system from functionalization, protein structure, macrophages, to biodistribution. I do appreciate that this would be a significant re-write, but think it would help the reader.

2) The figures are dense with most including Panels a-i. They are well-done, but some of the impact is lost in the density. I would encourage the authors to look for panels that could be moved to SI (for example, Fig. 3d) and separate results from schematics. (I also find the shadowing effect used with some of the structures unnecessary, but that is just taste.)

On a technical level;

1) p. 2 - I agree that albumin is the correct protein to study, but describing it as "almost always the dominant constituent of the protein corona" without refs is risky. One could easily argue that the complement proteins or Ig's dominate.

2) Targeting to the lungs is unexpected and deserves more discussion.

3) Flow cytometry is used to quantify macrophage uptake, but is it clear that the nanoparticles are really internalized into the macrophages? It appears that the particles could just be tightly adhered

to the macrophage cell surface.

4) Toxicity of NP treatments is included in Fig. S6 and although not statistically significant at low concentrations, there is a definite increase in toxicity with 75% at 1 mg/mL, which causes some concern that cell health matters.

5) The rate of cell uptake of nanoparticles as a function of electrostatics should either be better supported (beyond refs. 46 or 47) or just cut.

Author's Response to Reviewer #1:

The manuscript describes the effects of modification of nanoparticles surface on their clearance by macrophage subsets. Overall, it is well written, logical and easy to read. The experimental procedures are thorough and well described. To achieve their aims, the authors synthesised 3 'soft' nanoparticles comprised of functionalized PEG hydrophilic blocks. These particles varied in surface charge, although the OH- and MeO- modified particles were very similar.

The findings are robust, and represent a contribution to the field. Some of the results confirm previous reports, which could be better referenced. A major weakness of the study is the number of nanoparticles examined. It is too small to draw any generalisation, especially with regard to both soft and hard PEGylated nanoparticles synthesised from other materials. Some of the experimental conditions are not optimum (see below). The authors should be more inclusive with their referencing, especially for work that previously reported similar findings, albeit with different nanoparticles. The following comments are provided for consideration by the authors.

1. Introduction:

a. While albumin is often the most prominent protein in coronas, it is not necessarily the most important. Minor proteins and non-protein molecules can be important determinants of NP fate in vivo. This should at least be acknowledged by the authors.

We agree with the reviewer that while albumin is one prominent constituent of the protein corona, its high abundance does not make it the most important. We intended to communicate the point that by relative abundance, albumin represents a substantial fraction of the protein coating formed on nanocarriers in blood. We have updated the main text to clarify this and to acknowledge the point made by the reviewer. The following revised statement is now included in the introduction:

“For intravenously administered nanocarriers, serum albumin usually accounts for a significant fraction of the protein coating formed on the underlying nanocarrier chassis, as it is the highest concentration protein in the blood. Despite this high fractional composition within adsorbed protein layers, albumin is considered to be relatively inert, as albumin-coated liposomes have been shown to reduce nanocarrier recognition by macrophages to an extent that was comparable to PEGylated liposomes (PMID: 29357249). Many proteins of lower blood concentration, such as Factor XII (PMID: 2146350) and kininogen (PMID: 6695362), have higher surface activity and are linked to key biochemical responses to nanocarriers such as complement activation and thrombosis.”

b. I am unaware of anyone in the field who has championed the concept that albumin is responsible for the stealth effect of PEGylated NP. This statement should be referenced or removed.

We thank the reviewer for bringing this to our attention. We did not intend to phrase any statement that would attribute the stealth effects of PEGylated nanoparticles to the stealth effect of albumin. We meant to confer that in terms of adsorbed proteins, albumin is considered to be relatively inert, and by not eliciting detrimental biological responses and minimizing opsonization (PMID:

29357249), it contributes to stealth effects of nanocarriers to prolong circulation time (PMID: 15567509; PMID: 17000067). This effect would be evident with or without PEGylation. For example, Sato et al., demonstrated that coating liposomes with albumin rendered these nanocarriers inert and decreased macrophage recognition. Sato et al., concluded that the albumin-coated liposomes had lower uptake by macrophages that was similar to what is observed for PEGylated liposomes (PMID: 29357249).

We acknowledge that readers may interpret a statement made in the past version of the abstract in the way raised by reviewer #1. To address the reviewer's concern, we have re-phrased this sentence of the abstract to read as follows:

“Interestingly, certain common chemistries that have long been considered to convey stealth properties **were found to** denature albumin, promoting nanocarrier recognition by macrophage class A1 scavenger receptors and providing a means for their eventual removal from systemic circulation.”

We have also have revised the text on **pg. 3** of the manuscript to briefly discuss and cite the work of Sato et al.:

“Despite this high fractional composition within adsorbed protein layers, albumin is considered to be relatively inert, as albumin-coated liposomes have been shown to reduce nanocarrier recognition by macrophages to an extent that was comparable to PEGylated liposomes (PMID: 29357249).”

c. The author should distinguish between unfolded and denatured albumin. These are different processes with different outcomes. One is reversible while the second is not. Consequently, the interaction with receptors may be very different for these different states of the same protein.

The reviewer raises an interesting consideration and we acknowledge the manuscript must be updated to better describe our usage of the two terms. We use both terms, “unfolded albumin” and “denatured albumin”, to synonymously describe an event where natively folded albumin loses its three-dimensional structure in response to an experimental perturbation (for example, after introducing the protein to a suspension of OH PS or MeO PS). We attest that neither term (“unfolded” protein or “denatured” protein) is distinguishable on the basis of reversibility, and this stance is supported by the following papers relevant to the subject:

1. Jackson, W., and Brandts, J. (1970). Thermodynamics of Protein Denaturation. A Calorimetric Study of the Reversible Denaturation of Chymotrypsinogen and Conclusions Regarding the Accuracy of the Two-State Approximation. *Biochemistry* 9(11): 2294-2301.
2. Anfinsen, C. (1973). Principles that Govern the Folding of Protein Chains. *Science* 181 (4096): 223-230.
3. Wallevik, K. (1973). Reversible Denaturation of Human Serum Albumin by pH, Temperature, and Guanidine Hydrochloride Followed by Optical Rotation. *J. Biol. Chem* 248(8): 2650-2655.

4. Go, N. (1975). The theory of reversible denaturation of globular proteins. *International Journal of Peptide and Protein Research* 7(4): 313-323.

5. Lee, J., and Hirose, M. (1992). Partially folded state of the disulfide-reduced form of human serum albumin as an intermediate for reversible denaturation. *J. Biol. Chem* 267(21): 14753-14758.

6. Pico, G. (1997). Thermodynamic features of the thermal unfolding of human serum albumin. *International Journal of Biological Macromolecules* 20(1): 63-73.

Figure 4 of our manuscript demonstrates, using multiple orthogonal techniques, that albumin loses its native structure upon adsorption to OH PS and MeO PS. When adsorbed to these surfaces, the tryptophan residues of albumin unpack in a quantifiable way (**Fig. 4c**), a loss of its alpha-helix secondary structure is measurable by circular dichroism spectroscopy (**Fig. 4i**), and the three-dimensional fold unravels such that it is more susceptible to protease cleavage (**Fig. 4f**). These analyses demonstrate that albumin (once natively folded) partially loses its structure when adsorbed to these nanocarrier surfaces. And this loss of protein structure, or denaturation, proceeds through an unfolding process. Therefore, we put forth that it is appropriate to refer to albumin as either “denatured” or “unfolded” when adsorbed to OH PS or MeO PS. We have updated the main text to describe our usage of the two terms.

The text now reads as follows: (**pg. 12**)

“For clarity, we note here that since albumin is natively folded prior to introducing the protein to nanocarrier suspensions, any observed loss of structure will be referred to as an “unfolding” or “denaturation” event in the present work.”

d. Reference 15 does not mention class AI scavenger receptors. Similarly, reference 14 does not include experiments that distinguish the different subclasses of scavenger receptors. The author should carefully check the appropriateness of the references throughout the introduction.

We thank the reviewer for their inquiry into the choice of references cited in the introduction. We’ve parsed through the introduction and have double-checked the references. We agree updates were necessary for statements discussing gp18 and gp30, which are endothelial scavenger receptors. For the purpose of clarity in this review, we will list each reference mentioned by the reviewer below, and we discuss changes made in our updated manuscript to respond to their concerns.

Previous reference 15:

Schnitzer, J., Sung, A., Horvat, R., and Bravo, J. (1992). Preferential interaction of albumin-binding proteins, gp30 and gp18, with conformationally modified albumins. Presence in many cells and tissues with a possible role in catabolism. *J. Biol. Chem* 267: 24544-24553.

Class A1 scavenger receptors are membrane glycoproteins located at the cell surface. Gp18 and gp30 are cell-surface glycoproteins present on endothelial cells that bind to chemically modified albumin, but not natively folded albumin. For this reason, it has been proposed that these proteins act as novel scavenger receptors (see Schnitzer and Bravo, 1993 – PMID: 8463286), and they function to remove oxidized, misfolded, and/or modified albumin from the bloodstream.

We have revised the text of the manuscript to more generally discuss gp18 and gp30 as membrane glycoproteins that exhibit scavenging functionality. We have also updated the text with references that demonstrate SR-A1 is capable of recognizing conformationally-altered albumin (Zhang et al., 1993. J. Biol. Chem, PMID: 8383674) and chemically-modified forms of albumin (such as maleylated albumin) (PMID: 10089827; PMID: 31190821).

For example, on **pg. 12**, we now state the following in the revised version of the manuscript:
“Tissue-resident macrophages express high levels of SR-A1 (PMID: 18800157) and structurally-modified forms of albumin have been reported as non-canonical ligands for these receptors (PMID: 8383674; PMID: 10089827; PMID: 31190821).”

Previous reference 14:

Fleischer, C., and Payne, C. (2014). Secondary Structure of Corona Proteins Determines the Cell Surface Receptors Used by Nanoparticles. J Phys Chem B 118: 14017-14026.

We have revised the introduction to remove the statement regarding SR-A1 specificity since the receptor was not explicitly mentioned by the authors in that work.

On pg. 16:

We have further revised the text to more generally describe Fleischer and Payne’s usage of fucoidan to probe the recognition of denatured albumin on cationic polystyrene by scavenger receptors expressed by monkey kidney epithelial cells *in vitro*. Furthermore, we have added multiple references that demonstrate the use of fucoidan as a ligand with good specificity for SR-A1 (PMID: 218198; PMID: 11251301; PMID: 22282357), which provides more evidence that this ligand can specifically probe recognition events by macrophage SR-A1. The basic and applied research findings that demonstrate and/or review fucoidan specificity for SR-A1 are discussed elsewhere in our response to reviewer comments.

To further address the issue raised by the reviewer, we have updated **pg. 16** of the manuscript to read as follows:

“To test these hypotheses, we pre-adsorbed albumin to the surface of each of the three PS chemistries and assessed their uptake by macrophages pre-treated with either 1x PBS (negative control) or saturating concentrations of a scavenger receptor competitor, fucoidan (2.5 mg/mL) (Fleischer and Payne, 2014) (**Fig. 5a**). Fucoidan is a sulfated polysaccharide derived from brown-algae that binds to SR-A1 with high specificity (PMID: 218198; PMID: 11251301). SR-A1 (CD204) is highly expressed by RAW 264.7 macrophages (**Fig. 5b**), and fucoidan has been used as an SR-A1 competitor ligand in this cell type by other efforts seeking to develop peptides with high specificity toward SR-A1 (PMID: 22282357).”

e. The authors state that the role of scavenger receptors in recognising and clearing soft nanocarriers has not been examined to date. That is not correct, unless I misunderstand the meaning of a soft nanocarrier. Perhaps the authors could better define this for the reader.

We agree with the reviewer that it is important to clarify our use of the term “soft nanocarrier”. At the same time, we note here for completeness that it is not our goal to formally define what is versus what is not a “soft nanocarrier” with this work, as that lies far outside its scope and is more suitable for publication by a consortium of labs (rather than it is a single lab). The initial version of the manuscript simply used this language to point out that our study focuses on self-assembling vesicular nanocarriers that have aqueous lumens and high elasticity. This general type of nanocarrier is widely investigated for its use in drug delivery applications (for example, polymeric vesicles or liposomes), and these structures are very different than hard, solid core nanoparticles in their chemical and physical properties. Broad definitions of “soft nanocarrier” have been published and include deformable (low elastic modulus) nanostructures composed of organic materials. This is in contrast to hard nanoparticles, which are typically inorganic (metallic: silver, gold, etc) or of high modulus within the GPa range (PMID: 32254332; PMID: 24892186; PMID: 21170133), which can include solid core polymeric nanoparticles likely polystyrene. (PMID: 21170133; DOI: 10.1039/978178262521600001). Of note, the vast majority of research into nanoparticle scavenging has been published using hard nanoparticles (and mostly in vitro), and a link between albumin structure on nanocarriers and scavenging by SR-A1 has not been reported for soft nanocarriers to the best of our knowledge.

We have updated the main text to clarify our use of the term “soft nanocarrier”. Furthermore, we’ve characterized the elastic modulus of PEG-*b*-PPS polymersomes to complement our other extensive physicochemical characterization of these soft nanocarriers. This rheometric analysis demonstrates that PEG-*b*-PPS polymersomes have an elastic modulus of <10 Pa (please see **new Supplementary Fig. 5** in the updated Supplementary Materials document).

Pg. 2 of the manuscript now states the following:

“Soft polymeric nanocarriers, **broadly defined by their deformability and organic composition**, are versatile platforms for controlled and targeted delivery of therapeutic payloads in vivo.”

Pg. 4 of the manuscript now states the following:

“In this study, we employ PEG-*b*-PPS vesicles as model soft nanocarriers by varying their surface chemistry without modifying their nanostructure.”

“Our rheological characterization demonstrates PEG-*b*-PPS polymersomes to have an elastic modulus of <10 Pa (Supplementary Fig. 5a, b). In combination with their polymeric composition, this low modulus verifies the expected physicochemical properties of PEG-*b*-PPS polymersomes as soft nanoparticles (PMID: 21170133; DOI: 10.1039/9781782625216-00001), distinguishing them from hard solid-core nanoparticles such as gold, silica, and polystyrene that typically have an elastic modulus in the GPa range (PMID: 32254332; PMID: 24892186; PMID: 21170133).”

Furthermore, we have updated **pg. 3** of the manuscript to better specify that the focus of this investigation is to study the research questions in the context of a self-assembling nanocarrier of low elastic modulus, which is a highly relevant category of materials that includes self-assembling polymeric nanocarriers and liposomes. The text on **pg. 3** of the manuscript has been revised to clarify this point:

“We aimed to determine whether surface principles established using gold, silica, and polystyrene solid core hard nanoparticles for controlling adsorbed protein structure apply to the design of a self-assembled soft polymeric vesicle.”

Aside from these changes, we do want to make a point that solid core nanoparticles made up of inorganic materials such as iron oxide and silica, and polystyrene (polymeric) solid core nanoparticles, have been investigated for scavenger receptor recognition in a small number of studies *in vitro*. These particles are often considered as hard nanocarriers due to their high rigidity, which can be measured based on elastic or storage modulus. We discuss the numerous differences between those studies and our work at length both in the manuscript and in our responses to reviewer comments in this document (for example, please see our response to **comment 4b**).

Aside from the use of entirely different nanomaterial classes in those investigations (solid core nanoparticles), they did not demonstrate surface chemistry-mediated control over albumin conformation, nor did they assess consequences of SR-A1-mediated recognition of albumin-denaturing nanocarriers on their clearance and biodistribution *in vivo*. Demonstrating these things requires using a set of nanoparticles that have different surfaces that either denature albumin or stabilize its structure. And most importantly, establishing this albumin fold-sensing as a clearance mechanism with physiological relevance further requires cellular-level biodistribution studies *in vivo* using the aforementioned set of nanocarriers.

We uniquely demonstrate these relationships in our work and this information cannot be found in other publications. Therefore, these core findings should not be attributed to a small number of *in vitro* studies performed with solid core nanostructures. The class of materials used in those investigations and the specific research questions addressed by their experiments were entirely different.

In the instances where related research questions have been studied for other classes of materials, those studies did not demonstrate what we describe above, nor did they demonstrate the consequences of the recognition events on nanocarrier biodistribution *in vivo*. We further discuss these differences elsewhere in this document in our responses to other reviewer comments (for example, please see our more detailed comments provided in response to **comment 4b** of this document).

2. Methodology:

a. The poly dispersion index for each of nanoparticles is reasonable, but this does not align well with the average diameters shown in supplementary figure 4 where the standard deviation for the diameter is 30 to 50% of the mean. Does the supplementary figure show between batch variation or within batch variation? If it is between batch variation, then how was this accommodated between experiments?

We thank the reviewer for an excellent observation. The displayed DLS data is the variation in diameter within a batch. The reported number average mean \pm s.d. in **Supplementary Fig. 4** is

consistent with the mean and PDI reported in **Table 1**. The PDI is calculated as follows (where \bar{d} is the mean diameter and σ is the standard deviation):

$$PDI = \frac{\sigma}{\bar{d}}$$

If one uses the number-average mean and PDI reported in **Table 1**, solving for the standard deviation recovers the error bars in **Supplementary Fig. 4**. For example, for the Phos PS the diameter of 81.8 ± 33.7 nm corresponds to a PDI of 0.17. This is a monodisperse formulation of polymeric vesicles. We do not see issues with the reported values.

b. What effect did the dextran loading have on size/charge?

The reviewer raises an important question that we did not present data for in the initial version of the manuscript. We did not observe any significant change in the size or charge of the nanocarriers with dextran loading. To address this comment, we have added characterization data for Dex_{70 kDa}-TMR-loaded PS into a new table (now **Supplementary Table 3**). Overall, Dextran loading did not lead to substantial changes in the size or charge of the nanocarriers. The main text has been updated to acknowledge the points made in this response and to reference the new data that has been added as **Supplementary Table 3** in the updated version of the manuscript.

c. Protein coronas were formed in 10% FBS. There is ample evidence in the literature to show that the corona formed in 10% serum is very different to that formed in 100% serum. Why choose this concentration in the first place? How might it reflect the expected corona following administration of the NP to mice?

We acknowledge that the protein composition of the corona does differ depending on the concentration. In a separate study, we have analyzed the composition of human plasma proteins adsorbed to a library of nine PEG-*b*-PPS nanocarriers. That study used modern label-free quantitative proteomic techniques to demonstrate albumin is a dominant constituent of the protein corona formed on all PEG-*b*-PPS nanostructures (including Phos PS, OH PS, and MeO PS). We have posted a pre-print of this work to the bioRxiv (DOI: 10.1101/2020.09.02.280404).

The present manuscript is a structurally focused mechanistic study that examines surface chemistry-mediated changes to the conformation of albumin adsorbed to vesicular nanocarriers of different surface chemistries, and its consequence on macrophage SR-A1 recognition. We therefore used 10 mg/mL BSA solutions for the majority of the study. Only the gel presented in **Figure 3a** utilized 10% FBS, and this was simply included as an example to demonstrate the strong presence of the 66-67 kDa band for BSA when serum proteins adsorb to each nanostructure type.

To address this comment, we've moved this gel to the supplement (**Supplementary Fig. 10**) and we've updated the main text of the manuscript to cite the pre-print of our more recent proteomics investigation (Karabin and Vincent et al., 2020. bioRxiv; DOI: 10.1101/2020.09.02.280404).

3. Results:

a. Supplementary figure 6 shows that 1 mg per ml nanoparticles causes a significant decrease in viability. This is contrary to what is stated in the text.

We thank the reviewer for bringing this issue with the manuscript text to our attention. In **Supplementary Fig. 7**, we acknowledge that there is a decrease in the viability of cultured RAW 264.7 macrophages for Phos PS and OH PS at the highest concentration tested of 1 mg/mL. However, we note that the cell viability is still high, above 80%, for these conditions of high polymer concentrations that would not be observed *in vivo*. The concentration of 1.0 mg/mL is too high to be realistically observed *in vivo* and is simply used here in this assay to demonstrate an upper bound for toxicity studies. Based on mouse blood volume and the highest injection concentrations for PEG-*b*-PPS nanocarriers used in this work, a more realistic polymer concentration that cells would sample *in vivo* is less than 0.42 mg/mL.

We have revised the manuscript to acknowledge that good viability was observed within relevant concentration ranges below 1 mg/mL Phos PS and OH PS at a high polymer. This change is listed below:

“The Phos PS, OH PS, and MeO PS were all non-toxic to RAW 264.7 macrophages in the physiologically relevant polymer concentration range of 0.01 – 1.0 mg/mL (**Supplementary Fig. 7**).”

b. While proteins bind rapidly to most NP, the protein corona changes with time and protein concentration. Those proteins with high concentrations bind first but they are displaced over time by minor components that are lower in concentration but have a greater affinity for the NP. It is during this evolutionary process that most tightly bound proteins unfold as they spread across the surface of the NP. This raises the important question of whether the experiments were performed at equilibrium. Why was a 2 hr incubation chosen?

We thank the reviewer for their comment regarding the rationale that underlies our use of the 2 h time point in this study. In the main text, we mention that “Short time scales were of particular interest to us, since early interactions with phagocytes have major consequences on nanocarrier fate and overall performance in many applications.” In our *in vivo* studies, we demonstrate that >50% of each PS type has been cleared from the blood by 4 h following IV administration (**Fig. 2b**). This finding suggested to us that we must select an earlier timepoint (< 4 h) where these clearance events were taking place. Furthermore, to minimize confounding effects associated with the exchange of albumin, the timepoint must be one where the albumin-rich protein corona was reasonably stable.

The 2 h timepoint served this purpose, and is supported by a separate, concurrent investigation in our laboratory that used state-of-the-art proteomics techniques with high sensitivity. We recently completed a proteomic analysis of the composition of human plasma proteins that adsorb to a library of nine different PEG-*b*-PPS nanocarriers after incubation for 2 h and 24 h (Karabin & Vincent et al., 2020. bioRxiv. DOI: 10.1101/2020.09.02.280404). This analysis included the Phos PS, OH PS, and MeO PS nanocarriers (as well as other morphologies that are far outside the scope of the present albumin-focused work). The cited study used modern label-free proteomics techniques to demonstrate that serum albumin is a dominant constituent of each of the PS protein

coronas formed in pooled human plasma at 2 h and 24 h. This analysis demonstrates the high relative abundance of albumin in each PS corona (that develops by 2 h) remains stable by 24 h. This further supports our use of the 2 h timepoint, since it demonstrates that the protein corona formed on each PS structure at 2 h has a high concentration of serum albumin. And perhaps most importantly, that this albumin concentration has largely settled since substantial changes are not observed 22 h later at a 24 h time point.

To address the comment made by the reviewer, we have updated the main text to provide this rationale and to cite the pre-print of our detailed proteomics analysis completed as a separate research investigation (Karabin & Vincent et al., 2020. bioRxiv. DOI: 10.1101/2020.09.02.280404). The section titled “**Albumin adsorption to polymersomes increases their uptake by macrophages**” now reads as follows:

“Serum protein adsorption to each PS type results in the formation of a heterogeneous protein corona that is rich in serum albumin. Our *in vivo* studies demonstrated that >50% of each PS type has been cleared from the blood by 4 h following IV administration (**Fig. 2b**). This finding suggesting our more focused, mechanistic investigations at the molecular- and cellular-levels required an even earlier timepoint where these clearance events were occurring. And to avoid confounding effects arising from the (potential) dynamic exchange of albumin, this timepoint must be one where a reasonably stable albumin-rich protein corona has formed. Using a 2 h timepoint satisfied these criteria and is supported by our proteomic analysis of the composition of human plasma proteins adsorbed to a diverse library of PEG-*b*-PPS nanocarriers (Karabin & Vincent et al., 2020), where the relative abundance of albumin in each PS protein corona developed by 2 h remained stable by 24..”

c. Supplementary figure 7 clearly shows that the stability of the particles were not the same. However this is not what is stated in the text.

We thank the reviewer for pointing out this issue with the manuscript text. We have updated the main text to explicitly state that Dex-TMR retention was lower in cold storage for OH PS and MeO PS nanoparticles. The main text now reads as follows:

“Differences in the stability of cargo loading were observed after cold storage at 4 °C for six weeks (**Supplementary Fig. 8, a-c**). The anionic Phos PS exhibited the greatest retention of Dex-TMR cargo, whereas the storage stability of Dex-TMR-loaded OH PS and MeO PS was lower (**Supplementary Fig. 8, b-c**).”

d. In figure 5C, there was a significant decrease in cell viability. It is unclear why the authors have chosen 75% viability as a cut-off. This appears random.

We thank the reviewer for raising this point. In assays like this, horizontal lines of this nature are often arbitrary and are included to facilitate interpretation. We have removed this line from the viability plot referenced by the reviewer. This plot is now presented as **Supplementary Fig. 18** (this data was moved out of the main text in order to simplify **Figure 5** in response to a different reviewer comment). For consistency, we have also removed the horizontal line from the **Supplementary Fig. 7**.

Furthermore, we acknowledge there is a decrease in cell viability at the highest concentration of fucoidan tested. However, we note that at this concentration, ~85% of the cells were still viable. This is not considered toxic (for example, see PMID: 30692675 and PMID: 32154232). We have updated the manuscript to indicate this:

“Fucoidan was non-toxic to RAW 264.7 macrophages at the high concentration used to saturate the receptors, as ~85% of the cells remained viable after an 8 h incubation period (**Supplementary Fig. 18**).”

e. In figure 5e, flow cytometry does not distinguish internalised versus surface bound NP. This needs to be considered in both the interpretation of the results and their discussion.

We agree with the reviewer that flow cytometry alone technically does not distinguish between internalized nanoparticles versus surface-associated nanoparticles. For clarification, we note here that our cellular uptake studies utilize multiple washing steps before harvesting cells for flow cytometry, which minimizes any signal associated with nanoparticles bound to the cellular membrane. In general, these types of *in vitro* assays are a common tool employed by our laboratory to study the rate of internalization of PEG-*b*-PPS nanocarriers under specific conditions of interest, and we have rigorously vetted these procedures in prior publications (PMID: 29468812; DOI: 10.1038/s41467-020-18657-5; DOI: 10.1002/smll.202004205). We now cite these papers on **pg. 10** of the manuscript.

Aside from our previously published work, we performed an analysis of polymersome colocalization with lysosomal compartments. After being internalized by macrophages, PEG-*b*-PPS nanocarriers are transported to lysosomal compartments. Confocal microscopy performed on cells stained with lysotracker demonstrates clear colocalization between polymersomes and lysosomal compartments at 4 h after being administered to the cells (representative images are displayed in **Supplementary Fig. 14**). This result demonstrates the nanocarriers are being internalized by the cells. These results, together with evidence from our optimized *in vitro* assays that have been published previously, demonstrate that our nanocarrier uptake studies quantify fluorescence resulting from the internalization of dye-loaded nanocarriers.

In addition to adding **Supplementary Fig. 14**, we have also updated the main text to clarify these points. While the reviewer made this comment in response to **Fig. 5**, we have added this text to the section titled “Albumin adsorption to polymersomes increases their uptake by macrophages”. We reasoned it is most appropriate to add this new content here since this location is the first place where *in vitro* uptake studies are described in the manuscript.

f. Use chemical inhibitors to identify specific receptors involved in ligand uptake is problematic. None of these is a highly specific, and all of them have the potential to directly interact with the particle as opposed to the target receptor. The use of cells that do not express target receptors or where the target receptors have been knocked down with siRNA is a much better protocol for identifying specific receptors involved in uptake.

We thank the reviewer for their comments regarding our use of fucoidan as an SR-A1 competitor ligand and their suggestions for alternative approaches. Fucoidan is a well-established ligand of SR-A1 receptors (PMID: 218198; PMID: 11251301). Early studies demonstrated that IV administration of fucoidan at high saturating concentrations blocks uptake of ¹²⁵I-acetyl-LDL by mouse peritoneal macrophages (PMID: 218198), suggesting specificity for SR-A1. Since this time,

fucoïdan has been established as a competitor ligand of SR-A1 that is often used as a tool to study SR-A1 engagement of nanoparticles and other materials *in vitro* (PMID: 22101717; PMID: 30763772). For example, fucoïdan has been used as a specific SR-A1 inhibitor in cultured RAW264.7 macrophages in the development of a specific SR-A1 binding peptide (PMID: 22282357). Furthermore, fucoïdan is only an established ligand for class A scavenger receptors and class C scavenger receptors (see Table I of Peiser and Gordon, 2001; PMID: 11251301). However, class C scavenger receptors have only been described in *Drosophila melanogaster* and have not been identified in mammals (PMID: 28483986). Taken together, multiple lines of evidence in basic and applied research demonstrate fucoïdan exhibits good specificity for SR-A1 in macrophages.

The use of fucoïdan thereby offers a tool for studying SR-A1 recognition of nanocarriers without the need for genetic intervention. Genetic intervention can have its own confounding effects arising from transient receptor modulation, the use of transfection reagents that alter cell membranes (for bringing siRNA into the cell), etc. Performing additional experiments with those molecular biology tools, and addressing their associated confounding effects, is outside the scope of this study.

As we described in our response to **question 1-d and question 1-e** above,, we have updated the manuscript to better clarify our usage of fucoïdan as a SR-A1 competitor and its specificity (see **pg. 16**). This update also addresses the comment raised by the reviewer here:

“To test these hypotheses, we pre-adsorbed albumin to the surface of each of the three PS chemistries and assessed their uptake by macrophages pre-treated with either 1x PBS (negative control) or saturating concentrations of a scavenger receptor competitor, fucoïdan (2.5 mg/mL) (Fleischer and Payne, 2014) (**Fig. 5a**). Fucoïdan is a sulfated polysaccharide derived from brown-algae that binds to SR-A1 with high specificity (PMID: 218198; PMID: 11251301). SR-A1 (CD204) is highly expressed by RAW 264.7 macrophages (**Fig. 5b**), and fucoïdan has been used as an SR-A1 competitor ligand in this cell type by other efforts seeking to develop peptides with high specificity toward SR-A1 (PMID: 22282357).”

g. It should be acknowledged that RAW cells do not express the entire repertoire of scavenger receptors. Consequently, these cells can only be used to identify those receptors that they express. This should be mentioned somewhere in the manuscript.

We appreciate the reviewer’s suggestion. We have updated the manuscript to acknowledge that RAW 264.7 macrophages do not express the entire repertoire of scavenger receptors at high levels. We have also added a reference to a recent proteomic analysis of RAW macrophages that supports this point (Guo et al., 2015; PMID: 25504905). The main text has been updated to include the following statement:

“While RAW 264.7 macrophages do not express the entire repertoire of scavenger receptors at high levels (Guo et al., 2015), its high SR-A1 expression makes it a suitable macrophage cell line for studying the influence of this receptor on nanocarrier uptake *in vitro*.”

For the purpose of review, we note that the cited proteomic study by Guo et al., 2015 did not find a statistically significant difference in the expression of SR-A1 by RAW 264.7 macrophages versus bone marrow derived macrophages. This finding further supports our use of this macrophage

cell line to investigate the role of SR-A1 in nanocarrier uptake.

4. Discussion:

a. A series of three nanoparticles is not very robust and care should be taken in over extrapolating the results from such a small sample size.

One major goal of this study was to determine whether PEG could be terminated with simple chemical groups to modulate the conformation of adsorbed albumin, and ultimately, the SR-A1-mediated cellular interactions of the nanocarrier-albumin complexes. We chose a strategy of rational surface chemistry selection instead of using a massive nanocarrier library. The three surface chemistries allowed us to investigate these specific differences. Our results suggest that if the surface chemistry of a ~100 nm sphere partially denatures albumin, it triggers recognition of the nanocarrier-protein complex by macrophage SR-A1. The use of a large nanocarrier library is beyond the scope of this current work.

b. There is little doubt that SR-A1 receptors are responsible for the uptake of most NP, especially following serum protein binding. This is not new. The author should place their finding in context with the many other papers that have reported the same observation.

To the best of our knowledge, this is the first work that (i) demonstrates surface chemistry-mediated control over the conformation of albumin adsorbed to a model soft PEGylated nanocarrier, and/or (ii) establishes SR-A1-mediated sensing of adsorbed albumin as a clearance mechanism employed by macrophages that alters biodistribution *in vivo*. These are two major advances of our work that appeal to researchers working in basic and/or applied drug delivery areas. We ask that care is taken to avoid grouping these findings with other reports that involve SR-A1 recognition of denatured albumin in a much more narrow context *in vitro*.

Our initial version of the manuscript cites the work of Fleischer and Payne (PMID: 24779411) on albumin denatured on cationic polystyrene nanoparticles and its recognition by receptors on the surface of monkey kidney epithelial cells *in vitro*, and the work of Chad Mirkin's group on oligonucleotide-functionalized gold nanoparticles (PMID: 21070003; PMID: 23613589). Aside from these publications, the closest study that we could find to the more specific aims and research questions of our work was conducted by Mortimer et al., 2014 (PMID: 24617595).

Mortimer et al., performed a quality *in vitro* study that reports on a potential role of SR-A1 in recognizing denatured albumin adsorbed to the surface of layered silicate platelets/discs. That study used disc-shaped nanomaterials that, as described by the authors, "agglomerate" into nanostructures with an 85 nm diameter. That study showed this single type of material denatures albumin, and they use polyinosinic acid pre-treatments to conclude SR-A1 binds denatured albumin that adsorbed to those platelets *in vitro*. While the cited study set up the *possibility* that an SR-A1-mediated recognition event has the *potential* for clearing nanomaterials that denature albumin, our investigation answers many critical research questions that were not addressed by that work. Furthermore, our work establishes a key mechanism that sets the stage for rationally designing soft PEGylated nanocarriers with controllable clearance by macrophages. We will discuss these advances in greater detail below.

Mortimer et al., 2014 used a more esoteric class of nanostructure (stacked silicate discs) that deviate substantially from more clinically relevant spherical nanocarrier suspensions (often PEGylated spheres, such as PEGylated liposomes). Furthermore, the cited study did not demonstrate

surface chemistry-mediated control over the modulation of adsorbed albumin structure to examine and control SR-A1 mediated recognition of a nanomaterial by macrophages. This is a key finding of ours that provides a new strategy for controlling nanocarrier biodistribution that has never been reported for a soft nanocarrier (including PEGylated nanocarriers). Lastly, their studies were conducted entirely *in vitro* and did not demonstrate the consequences of SR-A1 recognition of denatured albumin *in vivo*, which is required to make conclusions regarding the consequences of this event on nanocarrier clearance in general, or on biodistribution at the organ- and cellular-levels. This is not a criticism of Mortimer et al., 2014, as conducting *in vivo* biodistribution studies may not have been in the scope of their work. However, we mention this here during the review process to avoid attributing a key finding to a paper that did not present biodistribution data for supporting a SR-A1-mediated clearance mechanism employed by macrophages *in vivo*.

To summarize, our study provides advances in two major areas. First, it establishes the use of simple surface functionalization of a PEGylated vesicle as a rational design strategy for controlling the structure of albumin adsorbed to soft nanocarriers. Second, we demonstrate the role of fold-stabilization or denaturation of an adsorbed protein as a physiologically-relevant mechanism for controlling the clearance of nanocarriers by macrophages *in vivo*.

We have revised the manuscript (pg. 15-16) to cite Mortimer et al., 2014, as it is a work that we respect and it holds relevance to our work despite the differences that we describe in detail above:

“Changes in albumin conformation results in recognition by macrophage SR-A1 (PMID: 8383674; PMID: 24617595), and structural changes decrease the half-life of free albumin in the cardiovascular system (PMID: 6721834). Scavenger receptors contribute to the endocytosis of nucleic acid-functionalized gold nanoparticles (PMID: 21070003; PMID: 23613589), charged polystyrene nanoparticles (PMID: 24779411), and layered silicate platelets (PMID: 24617595). For example, cationic amine (NH₂)-modified polystyrene nanoparticles (58 nm diameter) denatured albumin and became recognized by scavenger receptors expressed on the surface of monkey epithelial cells (PMID: 24779411). In contrast, anionic carboxylate (COOH)-modified polystyrene nanoparticles (60 nm diameter) preserved albumin structure and were not recognized by scavenger receptors in the cited study (PMID: 24779411).

While one study demonstrated layered silicate platelets/discs denature albumin and are bound by macrophage scavenger receptors *in vitro* (PMID: 24617595), the relationship between adsorbed albumin folding state and recognition by macrophage SR-A1 has not been examined for soft polymeric nanocarriers. Furthermore, SR-A1 mediated recognition of protein-denaturing soft nanocarrier surfaces has not yet been established as a surveillance mechanism utilized by innate immune cells.”

Author’s Response to Reviewer #2

In recent years it has become clear that one of the main challenges of nanomedicines is avoiding rapid clearance to the liver and spleen. Instead of reaching their target, the vast majority of nanomedicines are rapidly cleared from circulation making them completely ineffective. This process is mediated by a "corona" of proteins that adsorb on the surface of nanoparticles in the bloodstream.

Vincent and co-workers address this challenge with a comprehensive study nanoparticle surface functionalization, corona formation, macrophage uptake, and *in vivo* targeting. The manuscript stands out for two reasons: 1) a complete story from the chemistry of functionalization to *in vivo* experiments and 2) a detailed study of the structure of the adsorbed protein and how changes in structure lead to the observed biodistribution. Of specific importance to the nanomedicine community is the role of PEG, a standard functionalization to avoid clearance, in this process. Overall, it is a careful and important study that should be published.

There were two organizational aspects that I did find detracted from the results.

1) The organization of the paper seems chronological rather than results-directed. For example, the results move from functionalization to biodistribution to macrophage uptake to protein structure. I expect this is the order that experiments were carried out. I can imagine that the observed biodistribution led the team to investigate further ultimately examining protein structure. However I think there would be value in rearranging to order in terms of the complexity of the system from functionalization, protein structure, macrophages, to biodistribution. I do appreciate that this would be a significant re-write, but think it would help the reader.

We appreciate the reviewer's suggestion to consider revising the order that the content is presented in the manuscript. While we agree with the reviewer that there are benefits associated with revising the manuscript order to increase with complexity (i.e. starting from the molecular level experiments to the cell culture studies to the *in vivo* work), we believe that the current manuscript order is best. Our reasoning for this is that the current organization first demonstrates system-level differences in nanocarrier biodistribution, and carefully probes prominent molecular-level influences that contribute to those differences. As mentioned by the first reviewer, prior studies have shown a linkage between NP clearance and albumin denaturation, so we would like the focus to stay on our detailed analysis of this effect at the protein conformation level and how phosphate surfaces influenced the effect by maintaining albumin fold state. Therefore, we have kept the overall organization of the manuscript the same, however, we improve the clarity of the manuscript elsewhere (as described by our responses to the reviewer's other comments regarding overall manuscript organization below).

2) The figures are dense with most including Panels a-i. They are well-done, but some of the impact is lost in the density. I would encourage the authors to look for panels that could be moved to SI (for example, Fig. 3d) and separate results from schematics. (I also find the shadowing effect used with some of the structures unnecessary, but that is just taste.)

We thank the reviewer for their suggestions regarding the organization of the manuscript figures, and we agree that there are cases where the panels could be either simplified or rearranged to improve clarity. We address each aspect of the reviewer's comment below:

Figure simplification: **Figure 3** has been simplified by moving both the SDS-PAGE gel (previously **Figure 3a**) and the serum albumin structural alignment (previously **Figure 3d**) to the supplementary materials. The gel is now presented as **Supplementary Fig. 10**. This structural alignment is now **Supplementary Fig. 11**. Moving this content allowed us to increase the size of the illustration, the

PFAM domain map, and the electrostatic maps of serum albumin. Additionally, we've updated portions of the main text that refer to **Figure 3** to improve the overall clarity of that section.

Furthermore, **Figure 5** has been further simplified by moving the effect of fucoidan on cell viability to the supplement. This viability data is now presented in **Supplementary Fig. 18**.

Separation of results from schematics: Previously, **Figure 5b** of the manuscript contained microscopy data that was embedded within the illustration that graphically depicts our *in vitro* cellular uptake studies. In response to the reviewer's comment, we have updated **Figure 5** to separate the microscopy data (**panel b**) from the schematic presented in **Figure 5a**. **Figure 5b** is now located at the top right corner of the figure. The size of these microscopy images has been increased to improve visibility for readers. The illustrations presented in **Figure 5a** have also been revised to accommodate this rearrangement.

Lastly, we have removed the shadowing effect from all main text figures (**Figures 1-3**) with one exception: we still use a shadow-like object beneath the text "Limited Proteolysis" in **Figure 4d**. We think this shadow improves the aesthetic of that particular label. However, this is just our preference. But again, **Figures 1-3** have all been updated to remove the shadowing effect per the reviewer's suggestion.

On a technical level;

1) p. 2 - I agree that albumin is the correct protein to study, but describing it as "almost always the dominant constituent of the protein corona" without refs is risky. One could easily argue that the complement proteins or Ig's dominate.

We thank the reviewer for raising this point. We acknowledge that complement proteins (especially C3) and Ig's are highly abundant. In addition to Albumin, we agree that compositional studies often find C3 and various Ig's as the most abundant proteins found in the corona formed on nanomaterials in blood. We further acknowledge that our previous phrasing of albumin as "the" dominant constituent of the protein corona was likely too strong. We have revised both the abstract and the main text of the manuscript to state albumin is simply "one of the" dominant constituents of the protein corona, which is in agreement with our recent proteomic analyses that quantify the relative abundance of blood proteins adsorbed to nanocarrier surfaces.

The abstract has been revised to state the following: (**pg. 1**)

"Albumin is initially one of the dominant proteins to adsorb to nanocarrier surfaces, a process that is considered benign or beneficial by minimizing opsonization or inflammation."

The introduction has been revised to state: (**pg. 2-3**)

"For intravenously administered nanocarriers, serum albumin usually accounts for a significant fraction of the protein coating formed on the underlying nanocarrier chassis, as it is the highest concentration protein in the blood. Despite this high fractional composition within adsorbed protein layers, albumin is considered to be relatively inert, as albumin-coated liposomes have

been shown to reduce nanocarrier recognition by macrophages to an extent that was comparable to PEGylated liposomes (PMID: 29357249). Many proteins of lower blood concentration, such as Factor XII (PMID: 2146350) and kininogen (PMID: 6695362), have higher surface activity and are linked to key biochemical responses to nanocarriers such as complement activation and thrombosis.”

2) Targeting to the lungs is unexpected and deserves more discussion.

We agree with the reviewer that targeting to the lungs is unexpected. We have updated the discussion section of the manuscript to elaborate on this observation further. The following sentences have been added to **pg. 19**:

“... Aside from these observations, we also note that OH PS was taken up at high levels by various immune cell subpopulations in the lungs. This finding was surprising to us, and it may suggest this nanocarrier type may be a useful chassis for targeted drug delivery applications seeking to treat respiratory conditions. However, this particular finding appears to be independent of albumin denaturation, since passive targeting was not observed for the MeO PS that also denatured albumin to a certain extent in the biophysical analyses presented in this work.”

3) Flow cytometry is used to quantify macrophage uptake, but is it clear that the nanoparticles are really internalized into the macrophages? It appears that the particles could just be tightly adhered to the macrophage cell surface.

We thank the reviewer for requesting clarification on whether the nanocarriers are internalized or not. A similar comment was also made by reviewer #1. As described there, our laboratory commonly assesses nanocarrier uptake *in vitro* and *in vivo* under conditions of interest, and we have thoroughly vetted these procedures in several publications (for three examples, please see the main text and supplementary experiments in the following more recent papers: PMID: 29468812; DOI: 10.1038/s41467-020-18657-5; DOI: 10.1002/smll.202004205). We now cite these papers on **pg. 10** of the manuscript. Despite demonstrating nanocarrier internalization in various publications in the past, we have also added new data to this paper to further address the concern raised by the reviewer. We will re-state our description of this experiment here (since both reviewers asked the same question):

We agree with the reviewer that flow cytometry alone technically does not distinguish between internalized nanoparticles versus surface-associated nanoparticles. For clarification, we note here that our cellular uptake studies utilize multiple washing steps before harvesting cells for flow cytometry, which minimizes any signal associated with nanoparticles bound to the cellular membrane. In general, these types of *in vitro* assays are a common tool employed by our laboratory to study the rate of internalization of PEG-*b*-PPS nanocarriers under specific conditions of interest, and we have rigorously vetted these procedures in prior publications (PMID: 29468812; DOI: 10.1038/s41467-020-18657-5; DOI: 10.1002/smll.202004205). We now cite these papers on **pg. 10** of the manuscript.

Aside from our previously published work, we performed an analysis of polymersome colocalization with lysosomal compartments. After being internalized by macrophages, PEG-*b*-PPS nanocarriers are transported to lysosomal compartments. Confocal microscopy performed on cells

stained with lysotracker demonstrates clear colocalization between polymersomes and lysosomal compartments at 4 h after being administered to the cells (representative images are displayed in **Supplementary Fig. 14**). This result demonstrates the nanocarriers are being internalized by the cells. These results, together with evidence from our optimized *in vitro* assays that have been published previously, demonstrate that our nanocarrier uptake studies quantify fluorescence resulting from the internalization of dye-loaded nanocarriers.

In addition to adding **Supplementary Fig. 14**, we have also updated the main text to clarify these points. While the reviewer made this comment in response to **Fig. 5**, we have added this text to the section titled “Albumin adsorption to polymersomes increases their uptake by macrophages”. We reasoned it is most appropriate to add this new content here since this location is the first place where *in vitro* uptake studies are described in the manuscript.

4) Toxicity of NP treatments is included in Fig. S6 and although not statistically significant at low concentrations, there is a definite increase in toxicity with 75% at 1 mg/mL, which causes some concern that cell health matters.

The reviewer raises important considerations regarding potential toxicity concerns when the nanocarriers are administered to RAW macrophages at a concentration of 1 mg/mL *in vitro*. A similar point was raised by reviewer #1. We have included our answer to that question here as well:

We thank the reviewer for raising this point. In assays like this, horizontal lines of this nature are often arbitrary and are included to facilitate interpretation. We have removed this line from the plot in **Supplementary Fig. 18** (this viability data was moved out of the main text in order to simplify **Figure 5** in response to a different reviewer comment). For consistency, we have also removed the horizontal line from the **Supplementary Fig. 7**.

Furthermore, we acknowledge there is a decrease in cell viability at the highest concentration of fucoidan tested. However, we note that at this concentration ~85% of the cells were still viable. This is not considered toxic (for example, see PMID: 30692675 and PMID: 32154232). We have updated the manuscript to indicate this:

“Fucoidan was non-toxic to RAW 264.7 macrophages at the high concentration used to saturate the receptors, as ~85% of the cells remained viable after an 8 h incubation period (**Supplementary Fig. 18**).”

5) The rate of cell uptake of nanoparticles as a function of electrostatics should either be better supported (beyond refs. 46 or 47) or just cut.

We have added two additional references to the manuscript (PMID: 32316537; PMID: 20690607) to support the effect of nanoparticle charge on cell uptake. These references are now cited on **pg. 10** of the updated version of the manuscript.

REVIEWERS' COMMENTS

Reviewer #1 (Remarks to the Author):

I have carefully reviewed the comments from the authors and found they addressed all of my concerns adequately. The manuscript now reads better and is more accurate in its statements and conclusions.

Reviewer #2 (Remarks to the Author):

Revisions are acceptable.